# Chromatin network retards nucleoli coalescence

Yifeng Qi [1] & Bin Zhang [1]✉

Nuclear bodies are membraneless condensates that may form via liquid-liquid phase separation. The viscoelastic chromatin network could impact their stability and may hold the key for understanding experimental observations that defy predictions of classical theories. However, quantitative studies on the role of the chromatin network in phase separation have remained challenging. Using a diploid human genome model parameterized with chromosome conformation capture (Hi-C) data, we study the thermodynamics and kinetics of nucleoli formation. Dynamical simulations predict the formation of multiple droplets for nucleolar particles that experience specific interactions with nucleolus-associated domains (NADs). Coarsening dynamics, surface tension, and coalescence kinetics of the simulated droplets are all in quantitative agreement with experimental measurements for nucleoli. Free energy calculations further support that a two-droplet state, often observed for nucleoli in somatic cells, is metastable and separated from the single-droplet state with an entropic barrier. Our study suggests that nucleoli-chromatin interactions facilitate droplets' nucleation but hinder their coarsening due to the coupled motion between droplets and the chromatin network: as droplets coalesce, the chromatin network becomes increasingly constrained. Therefore, the chromatin network supports a nucleation and arrest mechanism to stabilize the multi-droplet state for nucleoli and possibly for other nuclear bodies.

[1] Department of Chemistry, Massachusetts Institute of Technology, Cambridge, MA 02139, USA. ✉email: binz@mit.edu

Nuclear bodies are pervasive in eukaryotic cells and play a diverse set of functions[1], including RNA metabolism, transcriptional regulation[2], genome organization[3], etc. They are membraneless structures that mainly consist of protein and RNA molecules[4]. Their lack of a lipid-rich barrier allows rapid exchange of components with the nucleoplasm in responses to environmental cues and stress signaling[5]. Nuclear bodies also effectively increase the local concentration of enzymes involved in particular functions to facilitate more efficient cellular reactions[6].

Increasing evidence supports that nuclear bodies function as biomolecular condensates formed via liquid-liquid phase separation (LLPS)[4,7–9]. They exhibit round morphologies and dynamic fluid properties[10,11]. Two nuclear bodies can fuse into larger condensates following growth kinetics with similar scaling behavior as that observed for simple liquids[12–14]. In addition, their assembly was shown to be concentration-dependent, and the coarsening and growth dynamics can be quantitatively modeled with classical theories of phase separation[15]. At the molecular level, detailed mechanistic models for LLPS are beginning to emerge as well. In particular, low complexity domains and intrinsically disordered regions are enriched in many of the proteins associated with nuclear bodies[16–20]. These features enable non-specific, multivalent interactions that drive the formation of dynamical condensates.

However, several observations of nuclear bodies appear to defy predictions from classical nucleation and phase separation theories. In particular, these theories predict the thermodynamic equilibrium to consist of a single condensate that minimizes the surface energy[21,22]. On the other hand, multiple nucleoli (~2–5) can stably coexist in the same nucleus[11,12,23–26], as can paraspeckles[27] and nuclear speckles[28]. The exact number of nuclear bodies is sensitive to various factors, including cell volume[29] and nuclear lamina composition[30,31]. It has been proposed that non-equilibrium activities can dynamically alter protein-protein interactions to stabilize the multi-droplet state[15,32–35]. In addition, the chromatin network may suppress droplet coarsening through mechanical frustration as well[36–38].

Nuclear bodies and chromatin are also known to form attractive interactions[27], further complicating phase separation inside the nucleus beyond mechanical stress. For example, the upstream binding factor (UBF), which is a DNA binding protein and a key component of nucleoli, is known to recognize ribosomal DNA (rDNA) repeats to seed the rapid formation of nucleoli after cell division[39]. Correspondingly, rDNA and other chromosome segments, which are collectively noted as nucleolus-association domains (NADs)[9,40,41], can be seen inside and adjacent to nucleoli[42]. Paraspeckles[27] and speckles[28,43] have been found in spatial proximity with chromatin as well. In addition to proteins, nuclear bodies can harbor non-coding RNA to contact chromatin either by recruiting intermediate protein molecules or by forming RNA-DNA duplex or triplex[44]. Since chromatin forms a viscoelastic network spanning the nucleus[13,45–49], its interactions with nuclear bodies could impact the thermodynamics and kinetics of phase separation.

We carry out molecular dynamics simulations to investigate nucleoli formation with a computational model that explicitly considers nucleolar particles, the chromatin network, and the interactions between the two. We represent the chromatin network using a diploid human genome model that provides explicit polymer configurations for individual chromosomes. Interactions within and among chromosomes are optimized based on chromosome conformation capture (Hi-C) experiments to ensure in vivo relevance. The simulated dynamical process of droplet growth, coarsening, and coalescence are in quantitative agreement with experimental measurements. Importantly, our simulations predict the formation of multiple droplets, much like the coexistence of several nucleoli seen in the nucleus. We show that a two-droplet state is metastable and separated from the single-droplet state with an entropic barrier. The barrier arises from the chromatin network, which becomes more constrained upon droplet coalescence. Nucleolar particle-chromatin interactions link the motion between the chromatin network and the droplets, and stronger interactions are shown to produce more droplets. Our study provides insight into the critical role of the chromatin network on the formation of nucleoli and nuclear bodies in general.

## Results

**Phase separation with chromatin network leads to multiple droplets.** Leveraging a recently introduced computational model for the diploid human genome[50,51], we studied the impact of the chromatin network on nucleoli formation. We modeled the genome at the one megabase (Mb) resolution as a collection of 46 chromosomes inside spherical confinement (Fig. 1). Each chromosome is represented as a string of beads, which can be assigned as one of three types, *A*, *B*, or *C*. *A* and *B* correspond to the two compartment types that contribute to the checkerboard patterns typically seen in Hi-C contact maps[52,53], and *C* marks centromeric regions. Interactions among the beads were optimized to reproduce various average contact probabilities determined from Hi-C experiments for GM12878 cells using the maximum entropy optimization algorithm[54,55] (see Supplementary Material). Because of the non-equilibrium nature of the system, these experimentally derived interactions and temperature represent

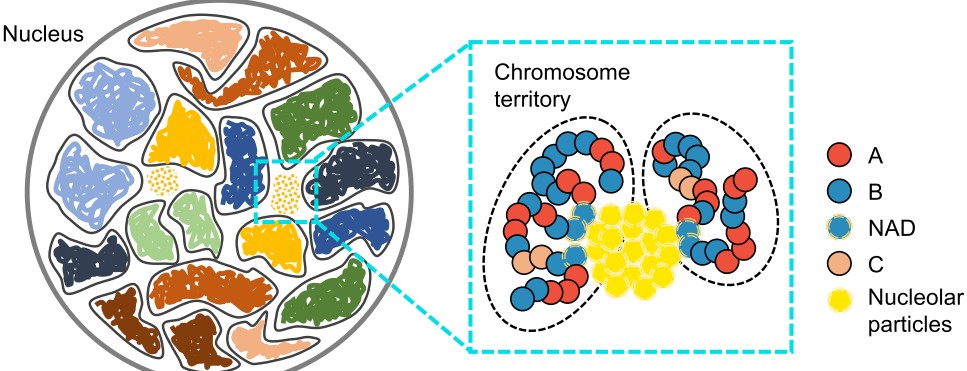

**Fig. 1 Overview of the computational model used for studying phase separation inside the nucleus.** The diploid human genome model represents each one of the 46 chromosomes as a string of beads confined in the nuclear envelope. Each bead is identified as compartment type *A*, *B*, or centromeric region *C*. Nucleolar particles share favorable interactions to promote phase separation and bind specifically with nucleolus-associated domains (NADs).

effective approximations to the steady-state distribution of genome organization (see "Methods"). While the model does not explicitly include histone modifications, transcription factors, or molecular motors, reproducing the contact probabilities between genomic segments measured in situ effectively allows it to account for the contribution of these factors to genome organization.

We introduced additional coarse-grained particles to model molecules that make up the nucleoli. The size and number of these particles were chosen based on the concentration of nucleolar proteins and the volume fraction of nucleoli (see "Methods"). Nucleolar particles share favorable interactions with each other and with nucleolus-associated domains (NADs), which are chromatin regions strongly associated with nucleoli[41]. A non-specific interaction term was also introduced between nucleolar particles and non-NAD chromatin regions. While our results are relatively robust with respect to the strength of these interactions (Supplementary Fig. 1), we found that, with the chosen values, the surface tension of simulated droplets compares favorably to experimental values for nucleoli (see Supplementary Material).

Starting from an equilibrated genome structure and randomly distributed nucleolar particles (see "Methods"), we carried out a total of twelve independent molecular dynamics simulations. These simulations lasted for 20 million steps, much longer than the relaxation timescale of chromosome conformations (Supplementary Fig. 2). In all but one case, the nucleolar particles aggregated into multiple droplets that persisted to the end of the simulations (Fig. 2b). This result contrasts with simulations performed without the chromatin network, where nucleolar particles always condense into a single droplet (Supplementary Fig. 3). Notably, the emergence

of the multi-droplet state is insensitive to the configuration used to initialize the simulations (Supplementary Fig. 4), the interactions between chromosomes (Supplementary Figs. 5, 6), and the resolution of the genome model (Supplementary Fig. 7). We found that nucleolar particles forming the droplets undergo dynamical exchange with the surrounding nucleoplasm while maintaining the droplet size (~1 μm) on timescales of several tens of minutes (Supplementary Fig. 8). Dynamical exchange of materials has been observed in fluorescence recovery after photobleaching (FRAP) experiments[56,57] and directly supports the liquid-like property of the droplets. The droplets are preferentially localized at the interior of the nucleus (Supplementary Fig. 9A). Because of their close association with these droplets, NADs are closer to the nuclear center than other heterochromatin regions as well[58]. However, not all NADs bind to the droplets, and a significant fraction of them localize towards the nuclear envelope (Supplementary Fig. 9B). Two classes of NADs that vary in nuclear localization have indeed been observed in prior studies[59].

Analyzing the simulated genome structures, we found that the model with nucleolar particles reproduces global features of the genome organization, including the formation of chromosome territories[60], the compartmentalization of heterochromatin/euchromatin[61], and the clustering of centromeric regions[62], as shown in Fig. 2c, d. The simulated chromosome radial positions agree well with experimental values[63], and the Pearson's correlation coefficient between the two is 0.89 (Fig. 2e). Because of its coarse resolution, the model will inevitably miss certain features of genome organizations, including the formation of chromatin loops and topologically associating domains[53]. While these structural motifs at fine scales are crucial for an accurate representation of the genome organization,

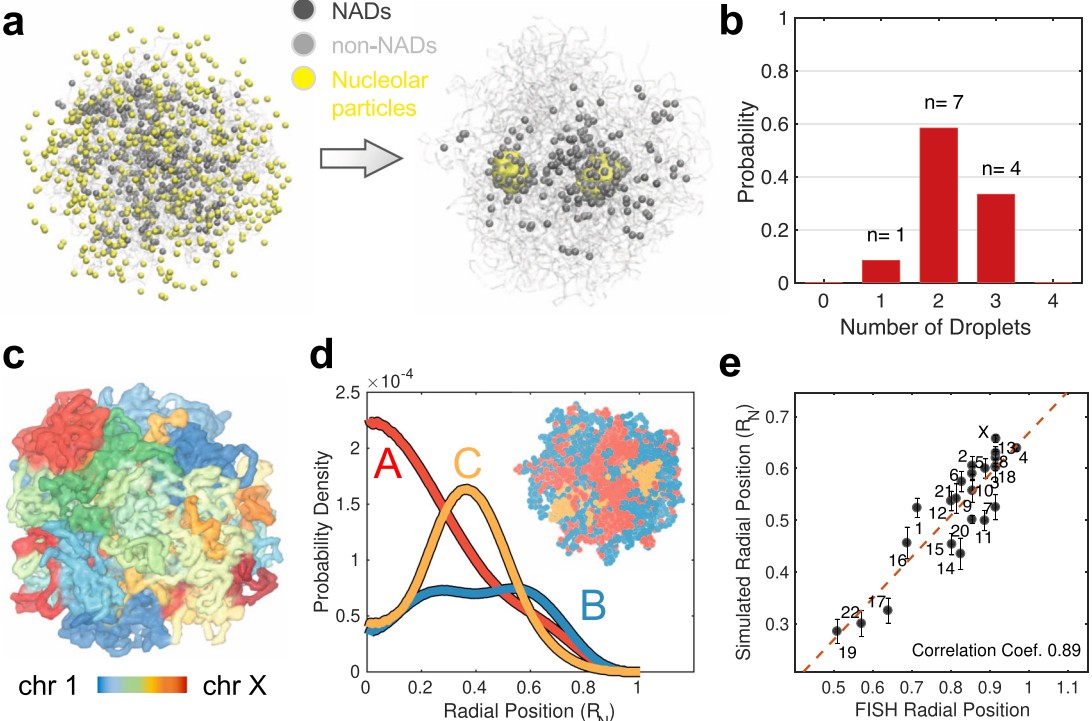

**Fig. 2 Multiple droplets form in dynamical simulations of the nucleus model. a** Representative initial (left) and final (right) configurations obtained from simulations, with nucleolar particles in yellow, NADs in black, and the rest of the genome in gray. **b** Probability distribution of the number of droplets observed at the end of simulation trajectories. **c** Representative configuration of the genome that illustrates the formation of chromosome territories. **d** Radial distributions of the different chromatin types that support their phase separation and preferential nuclear localization. An example genome configuration is shown as the inset, with the three types colored in red (*A*), blue (*B*), and orange (*C*), respectively. **e** Simulated radial chromosome positions correlate strongly with experimental values[63]. Error bars correspond to the standard deviation of the 12 mean values estimated using individual simulation trajectories. Homologous chromosomes were averaged together. $R_N$ is the radius of the nucleus used in polymer simulations.

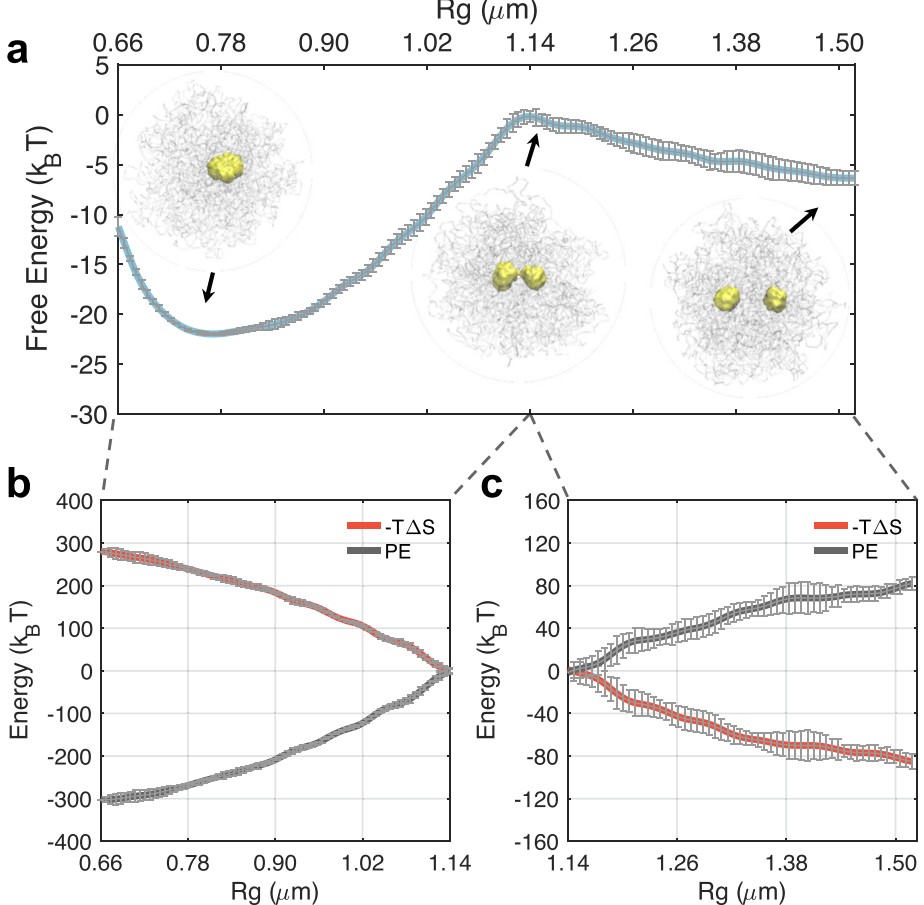

**Fig. 3 An entropic barrier hinders droplet coalescence and drives the metastability of a two-droplet state. a** Free energy profile as a function of the radius of gyration ($R_g$) that effectively measures the distance between two droplets. Energetic (black) and entropic (red) contributions to the free energy profile before (**c**) and after (**b**) the coalescence barrier. Error bars were calculated as standard deviations of the mean using block averaging by dividing the simulation trajectories into five blocks of equal length.

they are less significant for exploring the mechanisms of nuclear body formation, at least at a qualitative level (see Supplementary Fig. 7).

**An entropic barrier hinders droplet coalescence.** The emergence of a long-lived state with multiple droplets, while in contrast with predictions of the classical nucleation theory[21], is consistent with the coexistence of multiple nucleoli in the nucleus[11,13,24–26]. To better understand the stability of the multi-droplet state, we computed the free energy profile as a function of the radius of gyration ($R_g$) for a two-droplet system (see "Methods"). $R_g$ is defined as $\sqrt{\frac{1}{N}\sum_{i=1}^{N}\left|\mathbf{r}_i - \mathbf{r}_{\mathrm{com}}\right|^2}$, where $\mathbf{r}_i$ is the coordinate of the $i$th nucleolar particle and $\mathbf{r}_{\mathrm{com}}$ corresponds to the center of mass. The summation includes all $N$ nucleolar particles in either one of the two droplets. As the size of individual droplets remains stable, changes in $R_g$ will be mainly driven by variations in the distance of the two droplets. However, different from a simple center-of-mass distance, which becomes ill-defined if the lists of nucleolar particles participating in droplet formation are not updated on the fly, $R_g$ is relatively invariant with respect to the flux of particles between the two droplets (Supplementary Fig. 10). Simulations of the full system with both chromosomes and nucleolar particles were used to compute the free energy profile. Therefore, the impact of the chromatin network was accounted for implicitly even though it was not included in the definition of $R_g$. Umbrella sampling and temperature replica exchange were used to enhance conformational exploration.

As shown in Fig. 3a, the free energy profile exhibits two basins. While the basin at $R_g \approx 1.5 \,\mu m$ corresponds to the two droplet state observed before, an additional minimum with all nucleolar particles participating in a single droplet appears at smaller values ($\approx 0.75 \,\mu m$). The two basins are separated from each other with a transition state at $R_g \approx 1.13 \,\mu m$. Representative configurations at the transition state show a dumbbell shape with the establishment of a thin bridge between the two droplets. Consistent with predictions of the classical nucleation theory, the one droplet state remains as the global minimum. However, the merging of the droplets is kinetically constrained due to the presence of a barrier that is ~7 $k_B T$ in height. The barrier height is much larger than the error bars (0.5 $k_B T$) estimated via block averaging (see "Methods"), supporting its statistical significance.

To reveal the nature of the barrier, we decomposed the free energy into entropic and energetic contributions. Using the free energy profiles at different temperatures (Supplementary Fig. 11), we computed the entropy change along the collective variable with a finite difference method[64,65]. As shown in Fig. 3b, c, contributions from the entropy ($-T\Delta S$) continue to increase as $R_g$ decreases, and the droplets coalesce. While restricting the motion of the two droplets to smaller distances is naturally unfavorable, the entropic penalty is intensified here due to the increasingly restricted motion of chromosomes as well. As mentioned before, nucleolar particles form extensive contacts with NADs via specific interactions, and such contacts enforce correlative motions between droplets and chromosomes. The potential energy, on

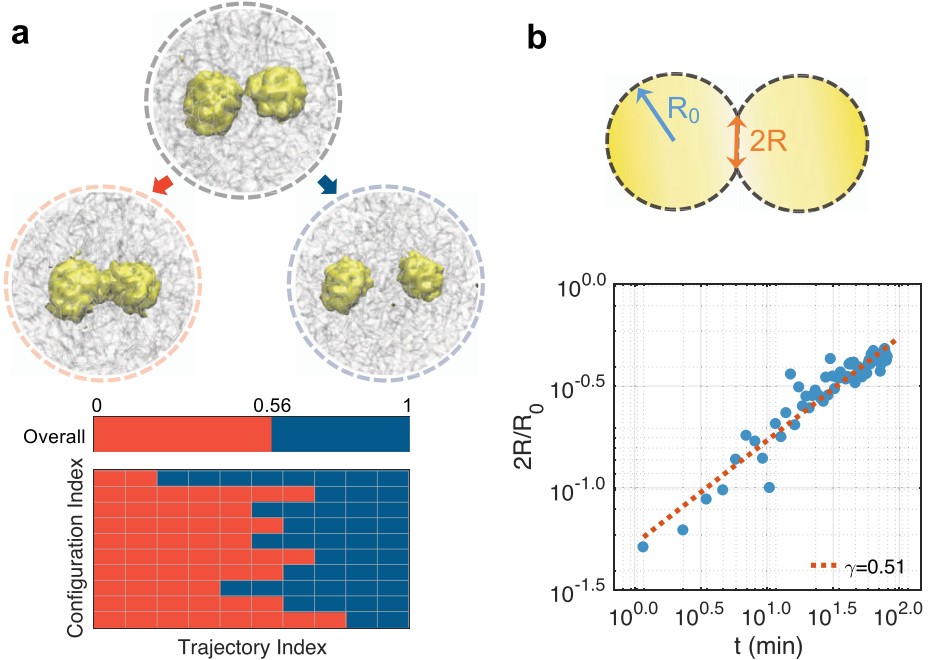

**Fig. 4 Dynamical characterization of droplet coalescence. a** Trajectories initialized from transition state configurations share approximately equal probability of committing to the two and single droplet state. The number of trajectories that ended up in the single droplet state is shown as red blocks in the bottom panel. Example configurations of the single-droplet (left), transition (middle), and two-droplet (right) state are shown on top. **b** Time evolution of the radius of the neck region between two droplets averaged over twelve independent simulations.

the other hand, favors droplet merging and decreases continuously along the collective variable due to the increase in nucleolar particle-nucleolar particle and nucleolar particle-chromatin contacts. Therefore, the transition state arises due to the presence of the chromatin network, and dissolving the polymeric topology of chromosomes indeed removes the barrier (Supplementary Fig. 3C).

While $R_g$ suffices for monitoring the progression of droplet coalescence, whether it serves as a "good" reaction coordinate or not requires further investigation. In particular, a good reaction coordinate should provide insight into the bottleneck that limits the reaction. Furthermore, trajectories initialized from the identified transition state will have an equal probability of committing to the reactant or product, or the so-called committor probability adopts a value of 0.5[66,67]. Otherwise, the transition state may bear little relevance to the reaction, and mechanisms derived from it can be misleading. To evaluate the significance of the transition state for droplet coalescence, we carried out additional simulations starting from random configurations with $R_g$ values at 1.13 μm. For each configuration, we initialized ten independent 200,000-step-long simulations with randomized velocities. We then counted the number of simulations that end up with the two-droplet state versus the single-droplet state. As shown in Fig. 4, among all these simulations, 56% of them led to the single-droplet state while the rest 44% ended up in the two-droplet state. These results strongly support the usefulness of $R_g$ for studying coalescence and the relevance of the identified transition state for mechanistic interpretation. Since the chromatin network was not included for defining the reaction coordinate and can vary significantly at a given value for $R_g$ (Supplementary Fig. 12), it may play a secondary or passive role in mediating coalescence.

We further isolated trajectories initialized from the transition state that led to the single-droplet state and computed the evolution of the neck radius as a function of time. As shown in Supplementary Fig. 13, the neck region was identified as the minimum of the density profile of nucleolar particles along the principal axis with the largest variance. As shown in Fig. 4b, by plotting the normalized neck radius ($2R(t)/R_0$) with respect to the time, we obtained a power-law relationship with exponent 0.51. $R_0$ is the average radius of the droplets before fusion. This exponent agrees with the experimental value determined for nucleoli[13] and suggests that droplet coalescence proceeds in the low Reynolds number regime dominated by viscous effects from the outer fluid, i.e., the nucleoplasm[68,69].

**Chromatin network gives rise to slow coarsening dynamics**. The thermodynamic analysis presented in Fig. 3 suggests that the chromatin network acts much like entropic springs to hinder the coalescence of droplets[70]. As the droplets move close to each other, they pull on the network and restrict the accessible polymer configurations. Restoring forces from the network to maximize configurational entropy counter droplet merging and give rise to the barrier. A similar mechanism could potentially impact the coarsening dynamics and the pathway leading to the formation of multiple droplets. To better understand the role of nucleolar particle-chromatin interactions in the overall phase separation process, we analyzed the dynamical trajectories at the onset of cluster formation.

We first monitored the time evolution of the number of clusters formed along the dynamical trajectories shown in Fig. 2. The clusters were identified as high-density regions of nucleolar particles across the entire nucleus using the DBSCAN (Density-Based Spatial Clustering of Applications with Noise) algorithm[71] (see Supplementary Materials). A typical trajectory is shown in Fig. 5a, and starts with zero clusters due to the random distribution of nucleolar particles in the initial configuration. The sudden appearance of nine clusters at time ~ 14 min suggests that nucleation can occur at multiple sites almost simultaneously. As time proceeds, the droplets began to merge or evaporate, and the trajectory eventually stabilizes to the two-droplet state.

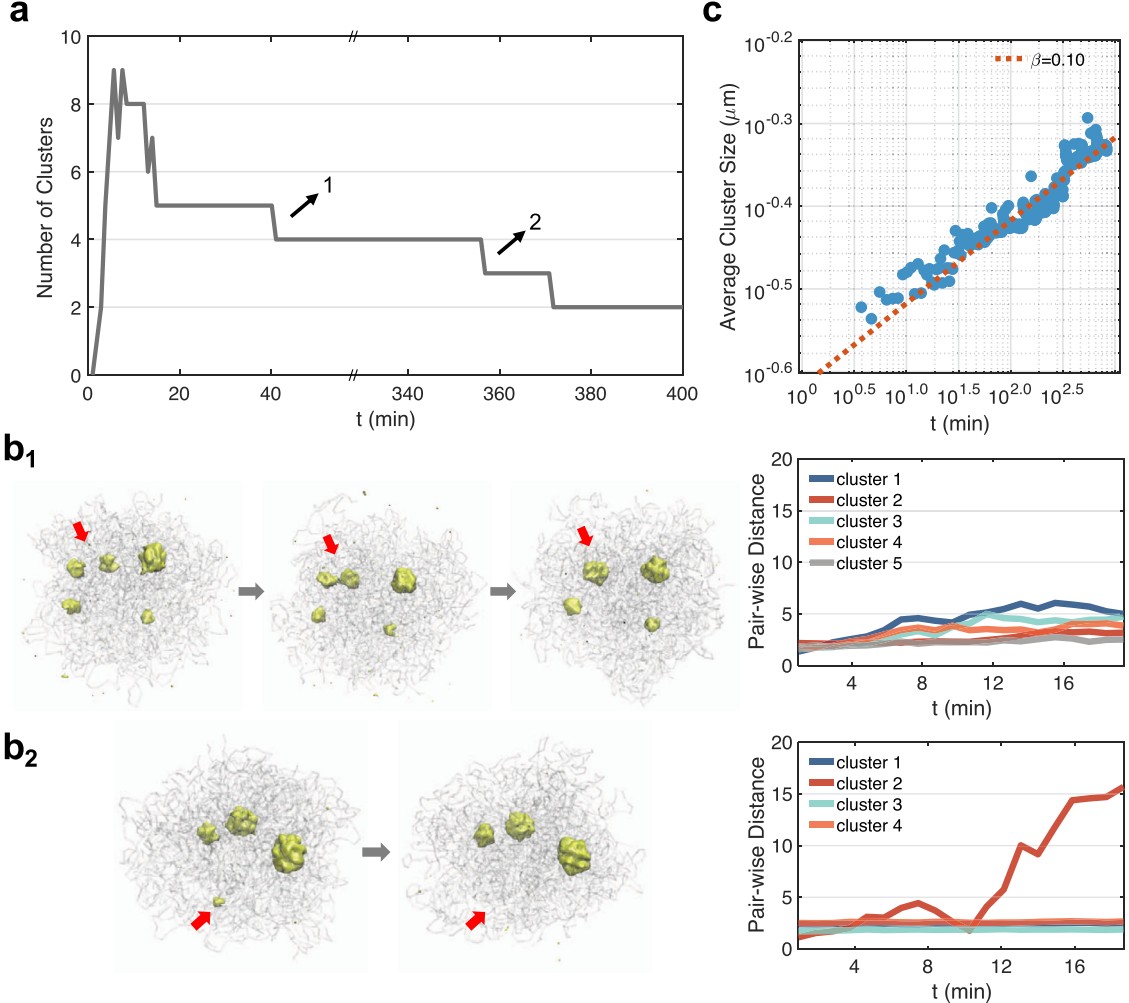

**Fig. 5 Coarsening dynamics and pathways for phase separation with the chromatin network. a** Time evolution of the number of clusters observed along a dynamical trajectory. **b** Detailed characterization of the two switching events that follow the Brownian motion-induced coalescence (**b**₁) and the diffusion-limited Ostwald ripening (**b**₂) path. The average pair-wise distance for each cluster remains relatively constant along the first path but increases significantly along the second one due to particle evaporation. **c** Power-law scaling of the average cluster size as a function of time.

We further recorded the average size of all clusters measured by their mean $R_g$. The increase of $R_g$ as a function of the time follows a power-law scaling with $R_g \propto t^\beta$. The exponent $\beta = 0.1$ differs from the phase separation process without the chromatin network. As shown in Supplementary Fig. 14, for simulations performed without the network, the initial cluster number is smaller and the exponent for cluster size growth is close to the theoretical value of $1/3$[72,73].

To provide insight into the appearance of the abnormal exponent $\beta$, we characterized the switching events that led to the decrease in cluster numbers. Specifically, we tracked the average pair-wise distance $\overline{r_{ij}(t)}$ of nucleolar particles within each cluster. In some cases, $\overline{r_{ij}(t)}$ for all clusters remain relatively constant and the switching occurs through the Brownian motion-induced coalescence (BMC) pathway (Fig. 5b₁). In other cases, we observe a significant increase in $\overline{r_{ij}(t)}$ before the switching occurs (Fig. 5b₂). This increase indicates an "evaporation" of the cluster following the diffusion-limited Ostwald ripening (DOR) path. We then identified all the switching events (42 in total) from the dynamical simulations and found that ~76% of them follow the BMC path. The rest of the switching events proceed via the DOR path and often involve smaller clusters to reduce the penalty associated with cluster disassembly (Supplementary Fig. 15).

The dominance of the BMC pathway explains the dramatic slowdown of the coarsening dynamics. In particular, the scaling exponent of $1/3$ was predicted based on a normal diffusion model in which distances between droplets scale linearly in time, i.e., $x^2(t) \propto Dt$[74]. However, as shown in Supplementary Fig. 16, most of the clusters exhibit sub-diffusion and $x^2(t) \propto Dt^{1/2}$. Assuming that the average size of droplets $r(t)$ is proportional to their mean distance, we arrive at $r(t) \propto t^{1/6}$ for the observed abnormal diffusion. The exponent now is closer to the value shown in Fig. 5c. The sub-diffusive motion arises from both the elastic stress produced by the viscoelastic chromatin network and the physical tethering of droplets to the chromosomes. In addition to the sub-diffusive motion, the chromatin network could further reduce the exponent and slow down the Brownian diffusion dominated coarsening dynamics by hindering droplet coalescence through entropic barriers similar to that shown in Fig. 3.

We note that the abnormal diffusion and slower coarsening kinetics have been directly observed by Lee et al. as well when monitoring the coarsening dynamics of model condensates based on intrinsically disordered protein regions in the nucleus[36]. In particular, they revealed a coarsening exponent of ~0.12, which is close to the value shown in Fig. 5c. The scaling exponent for nucleolar coarsening in vivo is also in good agreement with the simulated value when considering short time kinetics before 5 min[15].

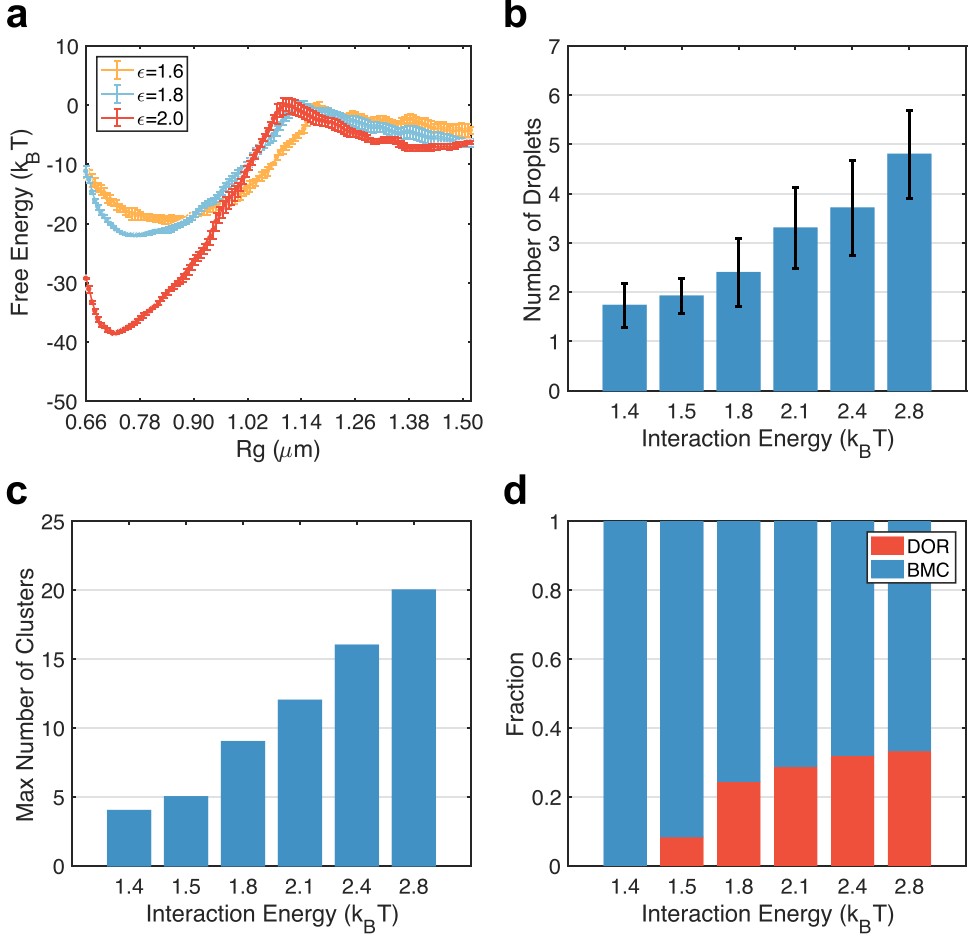

**Fig. 6 Nucleoli-chromatin interactions promote cluster nucleation. a** The free energy profiles of droplet coalescence at various nucleolar particle-chromatin interactions. The result for $\epsilon = 1.8$ is identical to the one presented in Fig. 3. Error bars were calculated as standard deviation of the mean using block averaging by dividing the simulation trajectories into five blocks of equal length. **b** Number of droplets formed at the end of 12 independent dynamical simulations at various strengths of nucleolar particle-chromatin interactions. Error bars correspond to standard deviations of results from independent trajectories. **c** The maximum number of clusters observed in dynamical trajectories performed at various nucleolar particle-chromatin interactions. **d** The fraction of cluster switching events following the Brownian motion-induced coalescence (BMC, blue) and the diffusion-limited Ostwald ripening (DOR, red) at various nucleolar particle-chromatin interactions.

**Nucleoli-chromatin interactions promote cluster nucleation.** Our results indicate that, while nucleolar particle-chromatin interactions increase the overall stability of the single-droplet state, they retard the coarsening kinetics by giving rise to sub-diffusion and entropic barriers. To more directly probe their impact on droplet coalescence, we recomputed the free energy profile at stronger ($\epsilon = 2.0\,k_BT$) and weaker ($\epsilon = 1.6\,k_BT$) nucleolar particle-chromatin interactions. As shown in Fig. 6a, while the stability of the merged state varied significantly, the transition and the two-droplet state are much less affected. These simulations again support the entropic origin of the free energy barrier.

We performed additional long-timescale simulations following the same protocol as those shown in Fig. 2. Starting from the same initial configurations, we varied nucleolar particle-chromatin interactions from $1.4\,k_BT$ to $2.8\,k_BT$ and carried out 12 independent 20-million-step long simulations for each interaction strength. We then computed the number of droplets formed at the end of these simulations. As shown in Fig. 6b, the number of droplets increases with the interaction strength. Stronger interactions facilitate the nucleation of nucleolar clusters on the chromatin, as can be seen from the increase of cluster numbers at the onset of phase separation (Fig. 6c). Cluster

coarsening, again, primarily follow the BMC pathway, though the ratio of DOR increases as well due to the instability of the nucleated clusters with a smaller number of nucleolar particles (Fig. 6d). Since merging of these clusters along the BMC pathway is hindered by entropic barriers, more nucleation naturally leads to increased droplet numbers at the end of the simulations. These results further support the role of the chromatin network in hindering droplet coalescence.

## Discussion

While nuclear bodies have been shown to exhibit liquid properties, the role of the surrounding environment on their formation and stability is less known. We modeled the dynamical process of phase separation that drives nucleoli formation in the presence of the chromatin network. The diploid genome model parameterized from Hi-C data was used to describe the interactions within and among chromosomes accurately. Simulations carried out with coarse-grained nucleolar particles succeeded in producing multiple droplets with dynamical behaviors comparable to nucleoli. We found that nucleoli-chromatin interactions facilitate the nucleation of condensates and retard their coalescence, stabilizing the multiple-droplet state.

Despite our best effort, the coarsened representations for the genome and nucleoli are bound to be approximate, and certain features of the biological system would inevitably be missed. While it may struggle at quantitative predictions, the model serves as a valuable tool for mechanistic explorations. In particular, since the nucleation and arrest mechanism arises from the generic polymer nature of the chromatin network, it should be insensitive to many of the model assumptions. Indeed, as shown in Supplementary Fig. 17, the multi-droplet state was also observed using a genome model parameterized with Hi-C data generated from clinical tissue samples with colorectal cancer[51]. The tumor Hi-C data and genome organization differ significantly from the one presented in the main text. The stability of the multi-droplet state is also preserved in additional perturbations that directly vary the resolution of the genome model (Supplementary Fig. 7), chromosome-chromosome interactions (Supplementary Fig. 5, 6), nucleolar particle size (Supplementary Fig. 18), and nucleolar particle-chromatin interactions (Supplementary Fig. 1).

Specific interactions between nucleolar particles and chromatin are crucial for the nucleation step and the emergence of the multi-droplet state (Supplementary Fig. 1). As such interactions may be present for other nuclear bodies, a similar mechanism could contribute to their formation as well. For example, several recent studies support the close contact between a subset of chromatin and speckles with distances at ~100 nm or less using techniques based on high throughput sequencing[75] or imaging[76]. Furthermore, these speckle-associated chromatin regions are largely conserved across cell-types[77], supporting the presence of non-random mechanisms for their maintenance. Close contacts with chromatin are likely stabilized by RNA molecules that are known to localize at speckle periphery[78,79]. Non-coding RNAs have also been found in paraspeckles and PML bodies to mediate their interactions with chromatin[27].

The mechanism uncovered here differs from the suppression of droplet growth in an elastic matrix discussed in several recent studies[36,37,80,81]. In contrast to our focus on the viscoelastic nature of the chromatin network and its attractive interactions with phase-separating agents, these studies were concerned with cross-linked gels and repulsive interactions. In the presence of permanent cross-linking, droplet-induced swelling of the polymer network can give rise to compressive stresses that arrest the coarsening dynamics completely. Therefore, the thermodynamic equilibrium becomes the monodisperse state with many droplets whose size is largely determined by the network mesh. Because of their favorable interactions in our model, there is substantial wetting of the chromatin network by nucleolar particles. The droplets incur minimal stress on the network (Supplementary Fig. 20), and the multi-droplet state remains metastable (Fig. 3). We note that the two mechanisms are not necessarily mutually exclusive and could contribute to the formation of different condensates inside the nucleus since the dynamical properties of chromatin are known to be timescale dependent[82].

Finally, it's worth mentioning that the nucleus is a non-equilibrium system, and active processes could contribute to the stability of the multi-droplet state as well[34,35]. For example, enzymes such as kinase could add post-translational modifications to condensate proteins and regulate their ability in engaging multivalent interactions. Explicitly modeling the active processes within the simulation framework outlined here may be necessary to account for the complementary mechanisms and provide a more comprehensive understanding of in vivo phase separation and nuclear body formation.

## Methods

### Details on the setup and simulations of the nucleus model.
We included both the genome and nucleolar proteins to simulate phase separation inside the nucleus.

Following the same setup and interactions as in ref. [50], we modeled all 46 chromosomes of the diploid human genome as beads on a string at the 1 MB resolution. In addition, a total of 500 particles were introduced to represent nucleolar proteins. See below for details on estimations for the particle size and number.

In addition to their attractive self-interactions, nucleolar particles bind with chromatin via specific and non-specific interactions. All three types of interactions were modeled with the cut and shifted Lennard-Jones potential

$$U_{LJ}(\mathbf{r}) = 4\epsilon \left[ \left( \frac{\sigma_l}{r_{ij}} \right)^{12} - \left( \frac{\sigma_l}{r_{ij}} \right)^{6} \right] + E_{cut} \qquad (1)$$

for $r_{ij} < r_c$ and zero otherwise, where $r_c = 2.0\ \sigma$. $E_{cut} = -4\epsilon[(\frac{\sigma_l}{r_c})^{12} - (\frac{\sigma_l}{r_c})^{6}]$. $\epsilon = 1.8$, 1.8, and 1.0 $k_B T$, and $\sigma_l = 0.5, 0.75$ and 0.75 $\sigma$ for nucleolar particle-nucleolar particle, nucleolar particle-NAD and nucleolar particle-non-NAD interactions. The values for $\sigma_l$ were chosen based on the size of nucleolar particles ($\sigma_P = 0.5\sigma$) and chromatin beads ($\sigma$) with arithmetic averaging. NADs were identified using the high-resolution sequencing data generated from ref. [41]. We processed the raw nucleolar-to-genomic ratios to generate signal data at the 1 Mb resolution. Only genomic loci with signals higher than a threshold value (15.0) were labeled as NADs and homologous chromosomes share identical NADs.

We used the software package LAMMPS[83] to perform molecular dynamics (MD) simulations in reduced units. Constant-temperature ($T = 1.0$ in reduced unit) simulations were carried out via the Langevin dynamics with a damping coefficient $\gamma = 10.0\ \tau_B$ and a simulation time step of $dt = 0.008\ \tau_B$, where $\tau_B$ is the Brownian time unit. Configurations were recorded at every 2000 simulation steps for analysis. The initial configuration of MD simulations was built as follows. We first obtained chromosome conformations from the end configuration of a 20-million-step long simulation of the genome-only model carried out in our previous study[50]. Next, 500 nucleolar particles were placed with random positions inside the spherical confinement. We further relaxed the system to avoid steric clashes by performing 400,000 step MD simulations. $\epsilon$ in Eq. (1) was set to 1.0 $k_B T$ for both specific and non-specific interactions to prevent cluster formation during the equilibration period. The last configuration of this equilibration simulation was then used to initialize the phase separation simulations. Analyses of simulation trajectories were carried out with in-house Python and MATLAB scripts.

We mapped the reduced units in MD simulations to physical units for direct comparison with experimental measurements. The length scale unit was determined by assuming a typical nucleus size with a radius of 5 μm. Since the spherical confinement mimicking the nucleus in our model adopts a radius of 19.7 $\sigma$, we have $\sigma = 0.254$ μm for the reduced length unit. We further determined the reduced time unit with the expression $\tau_B = \frac{30\pi\eta\sigma^3}{k_B T}$, which was obtained by matching the diffusion coefficient from simulations with that in the nucleus (see Supplementary Material). Using a value of the nucleoplasmic viscosity as $\eta = 10^{-2}$ Pa·s, we estimated $\tau_B = 3.6$ s.

We note that the energy unit ($k_B T$) in our model should be viewed as an effective temperature instead of strictly the biological value (310 K). Since the nucleus is a non-equilibrium system with constant perturbations from molecular motors and chemical reactions, the ensemble of genome organizations collected over a population of cells as in Hi-C experiments is unlikely to be in thermodynamic equilibrium. However, non-equilibrium distributions can often be well approximated by renormalized Boltzmann distributions with effective potentials and temperatures[84]. It is these effective quantities that we inferred from Hi-C contact maps to describe the interactions among chromosomes.

### Estimating the size and number of nucleolar particles.
The number of nucleolar particles used in simulations was selected based on the concentration of a representative nucleolar protein NPM1 as follows. For nucleoli of $2R_{Nu}$ in diameter, and a protein concentration of 1 μM[85], the number of nucleolar particles can be estimated as $N_p = \frac{4\pi}{3} \cdot N_A \cdot R_{Nu}^3 \cdot 1\ \mu M$, where $N_A$ is the Avogadro constant. We used 500 particles in simulations, and the corresponding $R_{Nu} = 0.58$ μm matches well with the observed size of nucleoli[11,13].

The size of the nucleolar particles can be determined assuming a space-filling model for the nucleoli as

$$\frac{\frac{4\pi}{3} \cdot \left( \frac{2^{\frac{1}{6}} \cdot \sigma_P}{2} \right)^3 \cdot N_P}{\frac{4\pi}{3} \cdot R_N^3} = \left( \frac{R_{Nu}}{R_N} \right)^3. \qquad (2)$$

Using $N_P = 500$, $R_N = 19.7$, $\sigma = 5$ μm and $R_{Nu} = 0.58$ μm, we have the diameter of nucleolar particles $\sigma_p = 0.5\ \sigma$. $2^{1/6}\sigma_p$ is the equilibrium distance between neighboring nucleolar particles in the Lennard–Jones potential.

We note that the above estimation is crude since many additional proteins and RNA molecules are present inside the nucleoli. Therefore, the nucleolar particles should be viewed as molecular aggregates with size $\sigma_P = 0.5\sigma - 0.12$ μm, rather than a single protein molecule. Given the size of a typical protein as 5–10 nm[86], the number of molecules within a single coarse-grained particle can be on the order of $10^3$. This number, while large, is, in fact, in the same order as the amount of distinct molecules that make up the nucleoli[87].

The molecular aggregate interpretation of nucleolar particles is also consistent with our use of $\epsilon = 1.8\ k_B T$ for specific interactions. This value appears much

smaller than the strength of typical non-bonded interactions, such as hydrogen bonds. $\epsilon$ should be viewed as "free energy" that accounts for the averaging over the heterogeneous interaction pattern of different molecules and the entropic penalty arising from confining individual molecules together. Furthermore, as stated above, the effective temperature in our model is presumably higher than 310 K, resulting in more significant values for $\epsilon$.

**Simulation details for free energy calculations**. We computed the free energy profile for coalescence using umbrella sampling and temperature replica exchange with 16 umbrella windows[88,89]. We defined the collective variable as $R_g = \sqrt{\frac{1}{N}\sum_{i=1}^{N} |\mathbf{r}_i - \mathbf{r}_{\text{com}}|^2}$. $\mathbf{r}_i$ is the Cartesian coordinate of the $i$-th nuclear particles found inside one of the two droplets and $\mathbf{r}_{\text{com}}$ is the center of mass. Indices of nucleolar particles in the two droplets were identified using the DBSCAN algorithm (see Supplementary Materials). A harmonic potential $\frac{K}{2} \cdot \left( R_g - R_c \right)^2$ was introduced in each umbrella window to facilitate the sampling of configurations at targeted distances. We chose the center of these windows, $R_c$, to be evenly spaced between 2.5 $\sigma$ and 6.0 $\sigma$ with an increment of 0.25 $\sigma$, except for the first window whose $R_c = 2.0 \, \sigma$. The spring constant $K$ was chosen as follows:

$$K = \begin{cases} 100.0 & 2.00 \le R_0 < 3.50 \\ 150.0 & 3.50 \le R_0 < 4.00 \\ 200.0 & R_0 \ge 4.00 \end{cases} \qquad (3)$$

Eight replicas were used within each umbrella window with temperatures ranging from 1.00 to 1.14 with 0.02 increments. Exchanges between these replicas were attempted at every 100-time steps. As shown in Supplementary Fig. 21, our choice of the temperature grid allows frequent exchange among replicas.

These simulations were initialized from a typical two-droplet configuration recorded at the end of a dynamical simulation and lasted twelve million steps. Configurations were recorded at every 400 steps. To compute the error bars and evaluate simulation convergence, we divided the simulation trajectories into five consecutive blocks with equal length. Free energy profiles were then calculated using only data collected from each block with the weighted histogram method (WHAM), and error bars were determined as the standard deviation of the five independent estimations.

**Reporting summary**. Further information on research design is available in the Nature Research Reporting Summary linked to this article.

## Data availability
The data that support this study are available from the corresponding author upon reasonable request. Simulation trajectories generated in this study have been deposited in the Zenodo database (https://doi.org/10.5281/zenodo.5570927). Source data are provided with this paper.

## Code availability
The computer code used in this study is available at https://github.com/ZhangGroup-MITChemistry/NucleoliCoalescence.

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

## Acknowledgements

We thank Dr. Xingcheng Lin, Dr. Kartik Kamat, and Dr. Xinqiang Ding for helpful discussions. This work was supported by the National Institutes of Health (Grant R35GM133580 to B.Z.) and by the Gordon and Betty Moore Foundation (Grant GBMF9162.15 to B.Z.).

## Author contributions

Y.Q. and B.Z. designed and performed the research, analyzed the data, and wrote the paper.

## Competing interests

The authors declare no competing interests.
