## [Peer Review File · Nature Communications]

REVIEWER COMMENTS

Reviewer #1 (Remarks to the Author):

The paper applied a previously developed genome folding model to study the thermodynamics and kinetics of droplet formation in the human nucleus. The authors performed extensive MD and enhanced sampling calculations that included not only chromatin fibre (from their previous studies) but also "protein" particles. It addresses an important and not well-understood topic on the influence of the nuclear environment and chromatin networks on nuclear body formation.

Since the results obtained in the paper are based on the model published elsewhere, I wonder how sensitive these findings are to the choice of the Hi-C data sets used to parameterize the model? In other words, how general are the main conclusions about the two-droplet state, enabling the nucleation and hindering of droplet coarsening. The spatial organization of chromatin is cell-type specific, and models may reproduce the population-averaged DNA-DNA ligation maps for one cell type, but not for another. Moreover, there is sizeable structural heterogeneity of chromatin within the same cell type - how does the current model reproduce the highly heterogeneous structural ensemble observed by high-resolution microscopy studies?

There are several polymer physics models that have been offered to describe chromatin folding. It might be worth mentioning the strengths and discuss the drawbacks of the authors' model. For example, specific interactions between different epigenetic histone marks, nucleic acids and proteins define the chromatin architecture, but it seems that they are not accounted for by this model. Resolution of 1Mb might not be high enough (loops are not modelled), since many Hi-C maps of higher resolution exist. However, I understand that increasing the resolution of the model might not be computationally plausible as of yet.

Comparisons with experimental data: figure 2e is rather impressive, but there are no error bars. In other places, comparisons are somewhat vague; statements like "consistent with Ref..." might not be enough. It is not clear how error bars are calculated on other figures - based on trajectories from different simulation runs?

Another comment is that the paper is missing essential details about simulations, Supplementary Information is pretty thin.

- How initial conformations of chromosomes were chosen in MD?
- "starting from an equilibrated genome structure", please provide more details.
- How replicas were constructed in REMD, temperature distribution and upper T limit were chosen, ensuring overlapping potential energies.
- How the convergence was checked for coarse-grained MD and REMD? It would be good to include the time evolution of replicas and/or dwell time.

Reviewer #2 (Remarks to the Author):

This manuscript titled "Chromatin Network Retards Droplet Coalescence" by Qi and Zhang presents coarse-grained molecular dynamics simulations of chromatin dynamics, liquid demixing and nucleolus formation in a whole cell nucleus. Their model for chromatin, described at the megabase level, was calibrated on experimental Hi-C data to reproduce chromosome organization in a previous article by the same authors. Here the focus is on the interplay between chromatin and liquid-liquid phase separation involved in the formation of nucleoli. To study this, the authors introduce coarse-grained

particles representing nucleolar proteins, describe their interaction with chromatin (including attractive interactions with nucleolar-associated domains), and observe their assembly into one or several droplets. They also use reweighted sampling to study the merging pathways between these droplets. Their main findings are: that the barrier to droplet merging is primarily of entropic nature; that these merging events dominate the dynamics, leading to slow coarsening.

Liquid-liquid phase separation inside cells, and in particular its interplay with the surrounding environment, is a subject of high current interest. This well-written theory article is original in the fact that it has the ambitious goal of representing a whole human cell nucleus in a semi-realistic way, with parameters calibrated on experimental data. The results are convincing and the conclusions appear well supported by the simulations, if not particularly striking. Overall, I have the feeling that this will be a useful addition to the literature - one that I will certainly cite - and that in the end it will deserve publication in Nature Communications. However, I do have some concerns that should be addressed first, as detailed below.

1) The balance between main text and SI is not appropriate in my opinion. Too many crucial details are relegated to the supplement, making the main text feel rather weak and not self-contained. I was unconvinced and judging this paper harshly after reading the main, only to find that most of my concerns had been addressed and discussed in the SI. This is in particular the case for the section "Estimating the Size and Number of Protein Particles" which would have its place in the main text.

2) Related to this SI section, I found the discussion of units and orders of magnitude rather weak in the main text. Many length scales are given without units; the size of the constituents cannot be found in the main text; as for time scale one has to dig deep into the SI to realize that they have actually been given serious consideration. Similarly, the level of coarse-graining is insufficiently discussed: as I understand, individual beads on the chromatin model represent 1Mb, i.e. huge molecular weight... but it is unclear in the main text whether the "protein" beads represent one or many proteins. It would be good to discuss more precisely what these proteins are and how many of them are represented in a single MD particle (given the size of 0.12 μ m indicated in the SI, it is a large number).

3) Still on the subject of dimensions and orders of magnitude, I was surprised to see energy scales of only $\sim kT$ involved in interactions at the Mb scale. One has to go to the SI to find mention that these are free energies with major interplay between entropy and energy. This deserves a serious discussion in the main text. Similarly, the mention of activity is only in the last sentences of the discussion. Should we really be thinking of kT as being the thermal energy here? Or is it an effective temperature including the nonequilibrium effects? How important are these? This deserves a more thorough discussion, and to be addressed earlier.

4) Regarding the discussion of the literature, there has been in the past few years a significant theoretical and experimental effort to understand the mechanical interactions between phase-separating droplets and surrounding *elastic* networks (see for instance: Cell 175, 1481 (2018); Physical Review X 8, 011028 (2018); Journal of the Mechanics and Physics of Solids 145, 104153 (2020); arXiv:2102.02787). Several of these works consider the importance of chromatin elasticity on nuclear condensate formation, in particular. How would these scenarios play out in the simulations presented in this paper? Would it be possible in these simulations to characterize the stresses induced by the droplets in the chromatin network, and the corresponding mechanical pressure this induces in the droplets? I understand that the authors are taking a different point of view here and that this might be beyond the scope of this paper, but can they at least relate their results and model to this body of literature?

Minor points:

- Fig 2E: what is the initial configuration in the simulations? Is this distribution of radial positions equilibrium, or does it just reflect the initial conditions?

- p6, the authors mention heterochromatin condensation. The current understanding of these domains is that they are intimately connected to the condensation of a protein, HP1a, into chromatin-attracting phase-separating droplets (Nature 547, 241 (2017)). Is this effect included in the model? How?

- The quantitative definition of R_g should come much, much earlier, given how central this is... I got pretty annoyed trying to guess it.

- p15, " $\sigma = 0.5, 0.75$ and 0.75σ for protein-protein, protein-NAD and protein-nonNAD interactions." Why these values? How did the authors determine them?

Reviewer #3 (Remarks to the Author):

In this work by Qi and Zhang, authors present a computational study that investigates coarsening of liquid droplets in the cell nucleus in the presence of chromatin polymer network. Their computational model specifically focuses on exploring the role of chromatin in the thermodynamics and kinetics of the phase separation, by which these droplets form. Authors suggest that that a two-droplet state is a metastable state and separated from a one-droplet state by an entropic barrier. They also propose the protein-chromatin interactions to facilitate the droplet nucleation, but to hinder the droplet coarsening due the correlated motion between droplets. The main message of the paper is that protein-chromatin interactions arrest the phase separation of droplets in a multi-droplet state.

This paper asks an interesting question about the role of chromatin network in the phase separation of nuclear liquid droplets and nicely reproduces previous experimental observations. However, it is not immediately clear what new physical or biological insights it reveals. It appears that authors have mainly focused on reproducing the experimentally observed phenomenology, but have missed out on an opportunity to report on the behavior of the chromatin network, which in opinion of this reviewer, is the novel part in this work. Yet, this part of the manuscript is presently rather underdeveloped. Therefore, I cannot recommend publication of this paper in Nature Communications in its present form.

Major issues:

1. The Introduction is weakly referenced, with only 26 references out of 90 across manuscript and supplements, and with seminal experimental observations, which should be prominent in the Introduction, only appearing in the later parts of the paper or not at all.

2. The title of the paper focuses on the role of chromatin in coalescence of nuclear droplets, yet the manuscript only briefly describes the genome. The description of the genome part of the model is quite minimalistic, mostly directing the reader to previous publications. For example, the interaction potentials are only listed, but not explicitly given, similar for the initial polymer configurations that were obtained from HiC data. These should be all explicitly listed/described in the supplements of the present paper.

3. Authors do not describe how was the polymeric model equilibrated and how was the equilibrium defined, which can critically impact the observed entropic barrier between the multi-droplet state and one-droplet state. The polymer dynamics in different stages of the equilibration can impact the observed droplet formation and thus change the observed outcome of the model. This should be explained in detail and controls should be provided.

4. Authors mention the viscoelastic nature of the genome playing a key role in the droplet coarsening, yet a physical explanation of this point is presently missing. The model explicitly accounts for specific

interactions between proteins, and proteins with chromatin, and their respective excluded volume in form of the Lenard-Jones potential. However, the choice of the size of particles will change these interactions, altering the effect of the polymer mesh size on the protein particles and hence the timescale of their phase separation and droplet formation. Similarly, changes in the polymer part of the model, for example, the bead size will effectively change the polymer meshsize. In addition, the interactions chosen for the monomers in the genome will impact polymer relaxation times, which will impact droplet formation. Could authors elaborate on this? How were the parameters chosen? Was a phase behavior of different values of interaction parameters, proteins sizes, etc explored and relevant timescales obtained?

5. On p. 9 authors state: "The thermodynamic analysis suggests that the chromatin network acts much like rubber with entropic springs to hinder the coalescence of droplets." Authors do not explain this analysis, nor do they provide reference. This should be elaborated upon and put into the context of the chromatin viscoelasticity impacting timescales and length scales of droplet coalescence.

6. The explanation for the choice of the interaction strengths for specific and non-specific interactions is missing. Such choice requires an optimization of interactions in form of a phase diagram. It would be helpful, if authors could provide this or otherwise motivate their choice.

7. The model assumes purely thermally driven interactions and equilibrium thermodynamics, yet there are several experimental observations that show direct involvement of active processes in life of nuclear bodies, their formation via liquid-liquid phase separation, interactions with the genome as well as the nucleoplasm in vivo (Berry et al, PNAS, 2015, Falahati et al, PNAS, 2017, Caragine et al, eLife, 2019, Kim et al, J. Cell. Sci., 2019). Moreover, in the last two of these studies a correlated motion of nuclear bodies (nucleoli and speckles) was observed in the context of such activity. How do authors reconcile the non-equilibrium nature of the system with their model? And how do their observed correlated motions of the droplets relate to those observed in vivo?

8. The explicit description of the free energy profile for the system containing both the genome and the droplets is not shown. So, it is not clear, how this entropic barrier connects to the role of chromatin in the system.

9. Authors are wording their work as if it generally applies to all nuclear bodies, but their model accounts for existence of NADs, allowing for protein-chromatin interactions, which are an exclusive feature of nucleoli. Hence, a generalization of this model to other nuclear bodies, which have no specific interactions with NADs (or other genomic sequences), is unclear.

Minor issue:

1. Fig. S4 A – there is twice a typo in the cartoon: "Large Radius of Gryation"

Title: Chromatin Network Retards Droplet Coalescence

Manuscript ID: NCOMMS-21-07939

Authors: Yifeng Qi and Bin Zhang

We want to thank all three reviewers for their insightful comments. We revised the manuscript extensively to improve its clarity and performed additional analyses to address the reviewers' concerns. We believe that we have successfully addressed all the comments, and the quality of our paper has significantly improved.

In the following, we provide point-to-point responses to all the reviewer comments.

RESPONSE TO REFEREE 1

Comment 0: *The paper applied a previously developed genome folding model to study the thermodynamics and kinetics of droplet formation in the human nucleus. The authors performed extensive MD and enhanced sampling calculations that included not only chromatin fibre (from their previous studies) but also “protein” particles. It addresses an important and not well-understood topic on the influence of the nuclear environment and chromatin networks on nuclear body formation.*

Response:

We appreciate the reviewer’s positive assessment of the paper, and we thank him/her for the detailed suggestions and comments.

Comment 1.1: *Since the results obtained in the paper are based on the model published elsewhere, I wonder how sensitive these findings are to the choice of the Hi-C data sets used to parameterize the model? In other words, how general are the main conclusions about the two-droplet state, enabling the nucleation and hindering of droplet coarsening. The spatial organization of chromatin is cell-type specific, and models may reproduce the population-averaged DNA–DNA ligation maps for one cell type, but not for another.*

Response:

We thank the reviewer for this insightful comment and agree that chromatin organization is cell type-specific. However, as we will show below, the mechanism uncovered in our study for the formation of multiple droplets, i.e., nucleation and arrested coarsening, is rather robust with respect to parameters used to model the chromatin network.

To validate the generality of our results and mechanisms with respect to cell types, we parameterized an independent diploid genome model to carry out additional simulations of phase separation. Hi-C data generated from clinical tissue samples with colorectal cancer were used for model parameterization [Johnstone et al. Cell 2020]. As shown in Fig. R1A and R1B, the average contact frequencies at the compartmental level calculated using tumor Hi-C data differ significantly from those determined using the GM12878 data, i.e., the one used in the main text. We applied the same algorithm introduced in [Qi et al. Biophys J, 2020], which was used to derive the genome model presented in the main text, to optimize the tumor genome model.

The 3D organization of the tumor genome differs significantly from that of GM12878 cells. While chromosomal territories are still clearly present (Fig. R1C), the preferential localization of B compartments at the nuclear periphery is lost (Fig. R1E). Furthermore, the radial positions of tumor chromosomes correlate poorly with that of GM12878 cells (Fig. R1D). The dramatic differences in the genome organization of the two cell types render the tumor genome model a good test system for validating the robustness of the mechanism of nucleoli formation.

With the newly optimized interactions for the tumor genome, we introduced nucleolar particles to the system. The number of nucleolar particles and the strength of nucleolar particle-nucleolar particle and nucleolar particle-chromatin interactions were kept the same as those used in the main text. Following the same protocol as in the main text, we performed 12 independent 20-million-step-long simulations to probe phase separation. The distribution for the number of droplets recorded at the end of these simulations is shown in Fig. R1F. The multi-droplet state is again favored. Notably, the average number of droplets observed in tumor simulations is higher than that for GM12878 cells. Cancer nuclei are indeed known to have higher nucleoli numbers [Farley et al. *Chromosoma* 2015; Derenzini et al. *Acta Histochem.* 2017; Weeks et al. *Cell. Mol. Life Sci.* 2019]. A possible reason for this increase could be due to the reduced inter-chromosome interactions in tumor cells (Fig. R1B). Chromosomes are less constrained with weaker interactions, and the entropic penalty for bringing them into close contact upon droplet coalescence is conceivably higher. *Therefore, the chromatin network for the tumor genome, while differing significantly from that for GM12878 cells, stabilizes the formation of multiple droplets. Furthermore, our model succeeds in capturing the qualitative change in nucleoli number upon tumorigenesis.*

As additional validations to the proposed mechanism, we carried out simulations in which the genome model for GM12878 cells was perturbed by removing inter-chromosomal interactions (Fig. R2) or by removing both intra- and inter-chromosomal interactions (Fig. R3). These perturbations again alter the genome organization dramatically, as evidenced by the changes in radial chromosome positions and radial distribution of A/B compartments. However, the distributions for the number of droplets obtained using the same simulation protocol as in the main text again support the stability of the multi-droplet state. Therefore, together with simulations for the tumor genome, these results suggest that the multiple-droplet state is a result of the polymer nature of the chromatin network and relatively insensitive to specific interactions among chromosomes.

The reason that the stability of the multi-droplet state is robust with respect to chromosome interactions is because of the nucleation and arrest mechanism discussed in the main text. The entropic barrier is expected for any polymeric network. As long as the nucleolar particle-chromatin interactions persist, the motion of the droplets will be coupled with the polymeric network. Their fusion will be arrested as a result of the entropic cost for confining the network when droplets move close to each other. Indeed, if nucleolar particle-chromatin interactions were removed and droplets cannot nucleate on the chromatin, simulations will always produce single droplets (Fig. R4).

We included these new results in the Supplementary Material and added a new paragraph in the *Discussion Section* of the manuscript. The relevant text is quoted below.

Despite our best effort, the coarsened representations for the genome and nucleoli are bound to be approximate, and certain features of the biological system would inevitably be missed. While it may struggle at quantitative predictions, the model serves as a valuable tool for mechanistic explorations. In particular, since the nucleation and arrest mechanism arises from the generic polymer nature of the chromatin network, it should be insensitive to many of the model assumptions. Indeed, as shown in Fig. S17, the multi-droplet state was also observed using a genome model parameterized with

Hi-C data generated from clinical tissue samples with colorectal cancer [cite]. The tumor Hi-C data and genome organization differ significantly from the one presented in the main text. The stability of the multi-droplet state is also preserved in additional perturbations that directly vary the resolution of the genome model (Fig. S7), chromosome-chromosome interactions (Figs. S5-S6), nucleolar particle size (Fig. S18), and nucleolar particle-chromatin interactions (Fig. S19).

Figure R1: **Tumor genome organization, while differs significantly from the one presented in the main text, supports the stability of the multi-droplet state.** For comparison, see Fig. 2 of the main text. (A, B) Average intra (A) and inter (B) chromosome contact probabilities between various compartments estimated using Hi-C data for GM12878 (blue) and tumor (red) cells. (C) Representative configuration of the genome that illustrates the formation of chromosome territories. (D) Correlation between chromosome radial positions obtained using the perturbed model and the original results presented in the main text. (E) Radial distributions of *A/B* compartments. An example genome configuration is shown as the inset, with the two compartments colored in red (*A*) and blue (*B*) respectively. (F) Probability distribution of the number of droplets observed at the end of simulation trajectories.

Figure R2: The genome organization obtained after removing Hi-C optimized inter-chromosome interactions, while differs significantly from the one presented in the main text, supports the stability of the multi-droplet state. We followed the same protocols as those used to produce Fig. 2 of the main text to carry out 12 additional independent 20-million-step-long simulations with α_{inter} defined in Eq. S8 and Eq. S10 of the Supplementary Material set to 0. (A) Representative configuration of the genome that illustrates the formation of chromosome territories. (B) Correlation between chromosome radial positions obtained using the perturbed model and the original results presented in the main text. (C) Radial distributions of A/B compartments. An example genome configuration is shown as the inset, with the two compartments colored in red (A) and blue (B) respectively. (D) Probability distribution of the number of droplets observed at the end of simulation trajectories.

Figure R3: The genome organization obtained after removing Hi-C optimized intra- and inter-chromosome interactions, while differs significantly from the one presented in the main text, supports the stability of the multi-droplet state. We followed the same protocols as those used to produce Fig. 2 of the main text to carry out 12 additional independent 20-million-step-long simulations with α_{inter} defined in Eq. S8 and Eq. S10 and α_{intra} defined in Eq. S6 all set to 0. (A) Representative configuration of the genome that illustrates the deformation of chromosome territories. (B) Correlation between chromosome radial positions obtained using simulations of the GM12878 and tumor genome model. (C) Radial distributions of A/B compartments. An example genome configuration is shown as the inset, with the two compartments colored in red (A) and blue (B) respectively. (D) Probability distribution of the number of droplets observed at the end of simulation trajectories.

Figure R4: **Removing nucleolar particle-chromatin interactions destabilizes the multi-droplet state.** To examine the dependence of simulation results on the interaction between nucleolar particles and chromatin, we set the specific interaction strength, i.e., ϵ defined in Eq. 1 of the main text, to $1.0 k_B T$. The nucleolar particle-nucleolar particle interaction and other parameters for interactions among chromosomes were kept the same. We followed the same protocols as those used to produce Fig. 2 of the main text to carry out 12 independent 20-million-step-long simulations for each system. The number of droplets formed at the end of each simulation was then recorded to compute the corresponding probability distributions for the original model (A), the model that removes Hi-C optimized inter-chromosome interactions (B) and that removes both Hi-C optimized intra- and inter-chromosome interactions (C).

Comment 1.2: *Moreover, there is sizeable structural heterogeneity of chromatin within the same cell type - how does the current model reproduce the highly heterogeneous structural ensemble observed by high-resolution microscopy studies?*

Response:

To evaluate the quality of the simulated genome structural heterogeneity, we compared simulation results with the data reported by Zhuang and coworkers [Su et al. Cell 2020]. Via a high-throughput imaging technique, the authors reported 3D coordinates of continuous genomic regions for the diploid genome, allowing direct comparison with simulated structures.

We focused the analysis on the radius of gyration and average distance towards nucleoli of individual chromosomes, and the pair-wise distance between chromosomes. As shown in Fig. R5, the mean simulation results correlate well with experimental measurements averaged over cells. In addition, we observed large-scale fluctuations of chromosome conformations, and the variance of R_g can reach 20% of the mean values. Similar variations can be seen for the other two metrics as well. These results support the structural heterogeneity in both chromosome structures and global genome organization.

While the mean simulated results correlate well with experimental measurements, there are differences in their absolute values. Furthermore, the correlations between simulated and experimental variances are less satisfactory. A possible caveat in our comparison is that we used a spherical shape to model the nuclear envelope. The experimental nuclei are more ellipsoidal, with the z-dimension significantly smaller than the other two (4 μm v.s. 20 μm). There is also a considerable variation in the nucleus size among the cells studied experimentally. The asymmetry in the nucleus shape and heterogeneity in nucleus size could give rise to larger pair-wise distances between chromosomes and more significant fluctuations. The flatter nuclei could potentially squeeze chromosomes to produce larger R_g but smaller variances. Considering the impact of nucleus shape on genome organization would be an important and exciting future direction.

We acknowledge the shortcomings of the model and now explicitly state them in the manuscript *Discussion Section*

Despite our best effort, the coarsened representations for the genome and nucleoli are bound to be approximate, and certain features of the biological system would inevitably be missed. While it may struggle at quantitative predictions, the model serves as a valuable tool for mechanistic explorations.

Figure R5: **Simulated genome structures recapitulate conformational heterogeneity observed in microscopy images.** (A) Correlation between simulated values and experimental results [Su et al. Cell 2020] for the mean and standard deviation (SD) of the radius of gyration (R_g) for individual chromosomes. As illustrated in the left cartoon, R_g measures the overall size of the chromosome, and its standard deviation provides a quantification on the fluctuation and heterogeneity of chromosome structures. (B) Correlation between simulated values and experimental results [Su et al. Cell 2020] for the mean and standard deviation (SD) of the average distance to nucleoli (d_{Nul}) for individual chromosomes. As illustrated in the left cartoon, d_{Nul} measures the relative position of chromosomes with respect to nuclear bodies, and its standard deviation provides a quantification on the fluctuation and heterogeneity of genome organization. (C) Correlation between simulated values and experimental results [Su et al. Cell 2020] for the mean and standard deviation (SD) of the average distance between pairs of chromosomes (PWD). As illustrated in the left cartoon, PWD measures the relative position of chromosomes with respect to each other, and its standard deviation provides a quantification on the fluctuation and heterogeneity of genome organization.

Comment 2: *There are several polymer physics models that have been offered to describe chromatin folding. It might be worth mentioning the strengths and discuss the drawbacks of the authors' model. For example, specific interactions between different epigenetic histone marks, nucleic acids and proteins define the chromatin architecture, but it seems that they are not accounted for by this model. Resolution of 1Mb might not be high enough (loops are not modelled), since many Hi-C maps of higher resolution exist. However, I understand that increasing the resolution of the model might not be computationally plausible as of yet.*

Response:

We thank the reviewer for this insightful comment, and agree that epigenetic marks, nucleic acids, and proteins contribute to chromosome organization. However, we want to point out that the impact of these factors on contacts between genomic segments should be captured by in situ Hi-C experiments. Since our model strives to reproduce Hi-C contact probabilities, it indeed accounts for the effect of histone modifications and protein molecules in shaping genome architecture, at least partially.

For example, our model reproduces the power-law scaling of Hi-C contact probability as a function of genomic separation (Fig. R6A). This power-law scaling exhibits an exponent of -1 , which differs from the -1.5 expected for an equilibrium globule. An exponent of -1 could be the result of loop extrusion by Cohesin molecules [Fudenberg et al. Cold Spring Harb. Symp. Quant. Biol. 2017]. Therefore, the impact of Cohesin molecules and loop extrusion can be effectively accounted for in our model without explicitly simulating the extrusion dynamics.

Furthermore, our model reproduces the phase separation of *A/B* compartments and the preferential localization of *B* compartments towards the nuclear envelope (Fig. 2D of the main text). Phase separation between the two compartments is evident in Hi-C data as checkboard patterns and can arise due to the enrichment of distinct histone modifications in corresponding genomic segments. For example, *A* compartments are known to exhibit activation marks such as H3K27ac and H3K4me3, while *B* compartments are enriched with H3K9me3 and H3K27me3. These marks can further recruit protein molecules such as HP1 and PRC1, which themselves have been shown to phase separate as well [Larson et al. Nature 2017, Plys et al. Genes Dev. 2019]. Therefore, our model effectively captures the impact of epigenetic marks and protein molecules in driving phase separation and genome organization.

Finally, our model quantitatively reproduces the average pairwise contact probability between chromosomes (Fig. R6B). Such non-random inter-chromosomal interactions likely arise from dedicated protein molecules and specific nucleotide sequences. They are crucial for reproducing the radial position of individual chromosomes.

We added the following text to the manuscript to highlight the strength of the model.

While the model does not explicitly include histone modifications, transcription factors, or molecular motors, reproducing the contact probabilities between genomic segments measured in situ effectively allows it to account for the contribution of these factors to genome organization.

Figure R6: Comparison between simulated and experimental pair-wise contact probabilities. (A) The average intra-chromosomal contact probabilities as a function of genomic separation estimated with simulated (red) and experimental (blue) contact maps. (B) Average contact probability between pairs of chromosomes computed with simulated contact maps correlate well with experimental results.

We agree that, at the current resolution, many fine-scale features of genome architecture cannot be captured, including loops, topologically associating domains, and enhancer-promoter contacts. In addition to increasing the model resolution, one might need to account for specific histone modifications and protein molecules, as in the string-binder model [Barbieri et al. PNAS 2012] and the loop extrusion model [Fudenberg et al. Cell Rep. 2016], etc. We added the following text to highlight the limitations of our model

Because of its coarse resolution, the model will inevitably miss certain features of genome organizations, including the formation of chromatin loops and topologically associating domains [cite]. While these structural motifs at fine scales are crucial for an accurate representation of the genome organization, they are less significant for exploring the mechanisms of nuclear body formation, at least at a qualitative level (see Fig. S7).

While these features are crucial for a comprehensive understanding of genome folding, we do not expect them to impact our conclusions on the mechanism of droplet formation qualitatively. As mentioned in our *Response to Comment 1*, the polymer nature of chromatin network is sufficient to stabilize the multiple-droplet state.

Nevertheless, we carried out additional simulations of phase separation using a genome model that is of 10 times higher resolution than the one used in the current manuscript. Each bead in the new genome model represents a 100 kb long genomic segment (Fig. R7A). Interactions among coarse-grained beads were again optimized based on the Hi-C data processed at 100 kb resolution. As shown in Fig. R7B, the model succeeds at reproducing fine-scale structural features of genome organization, including topologically associating domains. It also incorporates an additional compartment type, compartment *I*, that becomes apparent in high-resolution Hi-C data [Johnstone et al. Cell 2020]. We followed

Figure R7: A 100kb resolution genome model supports the stability of the multi-droplet state. (A) Overview of the 100kb resolution model. Similar to the model presented in the main text, chromosomes are represented as strings of coarse-grained beads, each one of which represents a genomic region of 100kb in length. Each bead was further labeled as compartment *A*, *B*, or *I*. As detailed in [Johnstone et al. Cell 2020], compartment *I* differs from both *A* and *B* with unique histone modification and DNA methylation patterns. The energy function of this model is similar to $U_{\text{genome}}(\mathbf{r})$ defined in the Supplementary Material. One notable difference is our distinction between interactions within individual topologically associating domains (TAD) and those across different TADs. Together with the model's higher resolution, this differentiation allows it to recapitulate high-resolution structural motifs of the genome seen in Hi-C data. Due to the computational cost with increased resolution, we only modeled a single copy of each chromosome, i.e., the haploid genome. The exact expression of the energy function can be found in [Johnstone et al. Cell 2020]. Parameters in the energy function were again derived using the maximum entropy optimization algorithm based on the Hi-C data for normal human colon tissue samples processed at the 100kb resolution. (B) Comparison between experimental (bottom left) and simulated contact map (top right) for a representative genomic region from chromosome 1 (27Mb to 57Mb). Dotted blocks along the diagonal are TAD boundaries determined using experimental data with the software TADbit (Serra et al. PLoS Comp. Bio. 2017). (C) Probability distribution of the number of droplets observed at the end of twelve simulation trajectories. Insets are representative snapshots for the two-droplet and the three-droplet states.

the same simulation protocols as in the main text to simulate phase separation with the optimized high-resolution model. We introduced the same number of nucleolar particles into the model as what we have in the 1Mb model, but scaled the size of each protein bead similarly according to the procedure in the *Methods: Estimating the Size and Number of Nucleolar Particles* Section in the main text. All the interaction strengths for protein-protein and protein-chromatin interactions were set to be the same as those in the 1Mb model. The multi-droplet state is favored in these simulations as well (Fig. R7C). Therefore, the

qualitative mechanism of phase separation, i.e., nucleation and coarsening arrest, is robust with respect to model resolution.

Comment 3: *Comparisons with experimental data: figure 2e is rather impressive, but there are no error bars. In other places, comparisons are somewhat vague; statements like “consistent with Ref. . .” might not be enough. It is not clear how error bars are calculated on other figures - based on trajectories from different simulation runs?*

Response:

We thank the reviewer for this helpful comment. We added error bars, which were calculated as standard deviations (SD) of the means, on Fig. 2E in the main text. SD was determined from the 12 average radial positions of each chromosome estimated from independent simulation trajectories.

Error bars in the free energy calculations (Fig. 3 and Fig. 6 in the main text) were similarly defined. In particular, we divided long simulation trajectories (twelve million steps) into five consecutive blocks with equal length. Free energy profiles were then calculated using only data collected from each block, and error bars were determined as SD of the five independent estimations. We added more detailed descriptions in the *Method Section* of the manuscript as quoted below

These simulations were initialized from a typical two-droplet configuration recorded at the end of a dynamical simulation and lasted twelve million steps. Configurations were recorded at every 400 steps. To compute the error bars and evaluate simulation convergence, we divided the simulation trajectories into five consecutive blocks with equal length. Free energy profiles were then calculated using only data collected from each block with the weighted histogram method (WHAM), and error bars were determined as the standard deviation of the five independent estimations.

We further revised the main text to be more specific about the comparisons. Specifically, we made the following changes:

OLD text

Two nuclear bodies can fuse into larger condensates following coalescence dynamics consistent with simple liquids [cite].

NEW text

Two nuclear bodies can fuse into larger condensates following coalescence dynamics with similar scaling behavior as that observed for simple liquids [cite].

OLD text

Consistent with observations from fluorescence recovery after photobleaching (FRAP) experiments [cite], these droplets exhibit liquid-like property. As shown in Fig. S2, protein particles forming the droplets undergo dynamic exchange with surrounding nucleoplasm while maintaining the droplet size ($\sim 1 \mu\text{m}$) on the ten-of-minutes timescale.

NEW text

We found that nucleolar particles forming the droplets undergo dynamical exchange with the surrounding nucleoplasm while maintaining the droplet size ($\sim 1 \mu\text{m}$) on timescales of several tens of minutes (Fig. S8). Dynamical exchange of materials has been observed in fluorescence recovery after photobleaching (FRAP) experiments [cite] and directly supports the liquid-like property of the droplets.

OLD text

*We note that the abnormal diffusion and the slower coarsening kinetics are consistent with *in vivo* experimental results performed by Brangwynne and coworkers [cite]. In particular, they revealed a coarsening exponent of ~ 0.12 , which is close to the value shown in Fig. 5C.*

NEW text

We note that the abnormal diffusion and the slower coarsening kinetics have been observed by Lee et al. as well when monitoring the coarsening dynamics of model condensates based on intrinsically disordered protein regions in the nucleus [cite]. In particular, they revealed a coarsening exponent of ~ 0.12 , which is close to the value shown in Fig. 5C.

Comment 4: *Another comment is that the paper is missing essential details about simulations, Supplementary Information is pretty thin.*

Response:

We have provided explicit expressions for the energy function of the genome model in the Supplementary Material. Further, we added more text in the *Methods Section* to explain the mapping of reduced units to physical units and provide details on building the initial configuration for simulation. As detailed below, we also performed analyses to demonstrate the convergence of MD simulations. A total of 10 new figures was added to the Supplementary Material, which now consists of 32 pages.

Figure R8: **Probability distribution of the number of droplets observed at the end of 12 additional simulation trajectories initialized with different starting configurations.**

Comment 5: *How initial conformations of chromosomes were chosen in MD?*

Response:

The initial conformations of chromosomes were chosen from the end configuration of a 20-million-step long simulation carried out in our previous study [Qi et al. Biophys. J. 2020]. We added the following text to the *Methods Section* of the main text to provide more details on building the initial configuration.

The initial configuration of MD simulations was built as follows. We first obtained chromosome conformations from the end configuration of a 20-million-step long simulation of the genome only model carried out in our previous study [cite]. Next, the 500 nucleolar particles were randomly placed inside the spherical confinement. We further relaxed the system to avoid steric clashes by performing 400,000 step MD simulations with ϵ in Eq. 1 set to 1.0 $k_B T$ for both specific and non-specific interactions. The last configuration of this equilibration simulation was then used to initialize all MD simulations.

The 12 simulations presented in Fig. 2B of the main text were initialized with the same configuration but different random velocities. We carried out 12 additional 20-million-step-long simulations to further evaluate the robustness of our results with respect to the initial configuration. These new simulations were initialized with different configurations prepared as following. We first collected 12 uncorrelated sets of chromosome conformations from a long simulation trajectory of the genome-only model at equal time intervals. For each set of conformations, we then introduced nucleolar particles and relaxed the resulting structures following the same procedure detailed in the *Methods Section* of the main text. As shown in Fig. R8, the distribution for the number of droplets recorded at the end of these simulations is quantitatively comparable to that shown in Fig 2B. Therefore, results presented in the main text are not sensitive to the initial conformations of

chromosomes.

Comment 6: *“starting from an equilibrated genome structure”, please provide more details.*

Response:

As mentioned in the *Response to Comment 4*, we provided more text in the *Methods Section* on building the initial configuration. We added a reference to the *Methods* in the sentence to point the readers towards the relevant text.

We note that all simulation files are available on the Github page (<https://github.com/ZhangGroup-MITChemistry/DropletCoalescence/>) to ensure reproducibility.

Comment 7: *How replicas were constructed in REMD, temperature distribution and upper T limit were chosen, ensuring overlapping potential energies.*

Response:

We thank the reviewer for this comment. We agree that replicas in REMD need to be constructed carefully to maximize sampling efficiency. However, to the best of our knowledge, there are no well-defined procedures for choosing the temperature spacing and upper-temperature limit. As a result, manual tuning is often required on a system-by-system basis. In addition, one usually has to balance sampling efficiency with computational cost. Larger spacing values reduce the number of replicas needed to reach a specific upper T limit and hence the computational cost. However, the rate for successful exchange among replicas, i.e., acceptance ratio, and the sampling efficiency will drop at larger spacing. Therefore, we adjusted the temperature spacing to achieve an average acceptance ratio of ~ 0.3 for exchanges between replicas (Fig. R9). This value of acceptance ratio is considered reasonable in the literature [Qi et al. *Methods Mol. Biol.* 2018].

The upper T limit was chosen as 1.14 to balancing sampling efficiency and the availability of computational resources. If the temperature becomes too high, the droplets may become unstable, resulting in configurations that are not significant at lower temperatures. We included a total of 8 replicas and 16 umbrella windows. Since every replica was simulated with 2 CPUs, 16 CPUs were used for each umbrella simulation. In sum, a total of 256 CPUs, a number close to all the resources available to the group, was needed to compute the free energy profile.

In Fig. R9, we plotted the acceptance ratio for all 16 umbrella windows as well as the overlapping of probability distributions of potential energy between neighboring windows. An average acceptance ratio (~ 0.3) in Fig. R9A and significant overlapping between neighboring windows (Fig. R9B) support the reasonable construction of replicas, which ensures an adequate sampling efficiency in our simulations.

Figure R9: **The temperature distribution used in replica simulations ensures sufficient overlap between replicas to guarantee frequent exchanges.** (A) Acceptance ratio for exchanges attempted between neighboring replicas in all 16 umbrella windows. Error bars correspond to the standard deviation of the mean. The black dash line indicates the average acceptance ratio over all umbrella windows. (B) Probability distribution for the potential energy at different temperatures for the umbrella window centered at $1.14 \mu m$ (4.5σ). The temperature varies from 1 to 1.14 with an increment of 0.02. High temperatures are indicated in red and low temperatures in blue. Distributions for other umbrella windows show similar overlaps.

Comment 8: *How the convergence was checked for coarse-grained MD and REMD? It would be good to include the time evolution of replicas and/or dwell time.*

Response:

The convergence of simulations was gauged by the magnitude of error bars with respect to features of free energy profiles. In particular, we divided long simulation trajectories (twelve million steps) into five consecutive blocks with equal length. Free energy profiles were then calculated using only data collected from each block, and error bars were determined as the standard deviation of the five independent estimations of the free energy profile. As shown in Fig. 3 and Fig. 6 of the main text, the error bars are on the order of $\sim 0.5 k_B T$, much smaller than the free energy barrier ($\sim 7 k_B T$).

As suggested by the reviewer, to further validate the convergence of the simulations,

Figure R10: **Frequent exchanges between replicas were observed during molecular dynamics simulations.** (A) Temperature ID assigned to each one of the eight replicas as a function of simulation time. The temperature varies from 1 to 1.14 with an increment of 0.02 from ID 1 to 8. (B) The average dwell time at various temperatures for each replica. The black line corresponds to the average across replicas with errorbars indicating standard deviations.

we computed the time evolution of replicas in the temperature space. As shown in Fig. R10A, all replicas exhibit fast dynamics and can traverse the entire range of temperatures during the simulations. The estimated dwell time at each temperature is quite uniform (Fig. R10B). These results again support frequent exchange among replicas and the convergence of replica-exchange simulations. We included Fig. R10 as Fig. S21 in the Supplementary Material.

RESPONSE TO REFEREE 2

Comment 0: *This manuscript titled “Chromatin Network Retards Droplet Coalescence” by Qi and Zhang presents coarse-grained molecular dynamics simulations of chromatin dynamics, liquid demixing and nucleolus formation in a whole cell nucleus. Their model for chromatin, described at the megabase level, was calibrated on experimental Hi-C data to reproduce chromosome organization in a previous article by the same authors. Here the focus is on the interplay between chromatin and liquid-liquid phase separation involved in the formation of nucleoli. To study this, the authors introduce coarse-grained particles representing nucleolar proteins, describe their interaction with chromatin (including attractive interactions with nucleolar-associated domains), and observe their assembly into one or several droplets. They also use reweighted sampling to study the merging pathways between these droplets. Their main findings are: that the barrier to droplet merging is primarily of entropic nature; that these merging events dominate the dynamics, leading to slow coarsening.*

Liquid-liquid phase separation inside cells, and in particular its interplay with the surrounding environment, is a subject of high current interest. This well-written theory article is original in the fact that it has the ambitious goal of representing a whole human cell nucleus in a semi-realistic way, with parameters calibrated on experimental data. The results are convincing and the conclusions appear well supported by the simulations, if not particularly striking. Overall, I have the feeling that this will be a useful addition to the literature - one that I will certainly cite - and that in the end it will deserve publication in Nature Communications. However, I do have some concerns that should be addressed first, as detailed below.

Response:

We appreciate the reviewer’s strong assessment of the significance, novelty, and broad interest of the paper, and we thank he/she for the detailed suggestions and comments to improve the paper’s clarity.

Comment 1: *The balance between main text and SI is not appropriate in my opinion. Too many crucial details are relegated to the supplement, making the main text feel rather weak and not self-contained. I was unconvinced and judging this paper harshly after reading the main, only to find that most of my concerns had been addressed and discussed in the SI. This is in particular the case for the section “Estimating the Size and Number of Protein Particles” which would have its place in the main text.*

Response:

We thank the reviewer for this helpful comment. We have moved the section “Estimating the Size and Number of Protein Particles” from SI to the *Methods Section* of the main text.

In addition, following the reviewer’s suggestion, we added more text in the *Methods Section* to explain the mapping of reduced units to physical units, provide details on building the initial configuration for simulations, and comment on the non-equilibrium nature of the model interactions and temperature. The relevant text is quoted below

The initial configuration of MD simulations was built as follows. We first obtained chromosome conformations from the end configuration of a 20-million-step long simulation of the genome-only model carried out in our previous study [cite]. Next, 500 nucleolar particles were placed with random positions inside the spherical confinement. We further relaxed the system to avoid steric clashes by performing 400,000 step MD simulations. ϵ in Eq. 1 was set to $1.0 k_B T$ for both specific and non-specific interactions to prevent cluster formation during the equilibration period. The last configuration of this equilibration simulation was then used to initialize the phase separation simulations.

We mapped the reduced units in MD simulations to physical units for direct comparison with experimental measurements. The length scale unit was determined by assuming a typical nucleus size with a radius of $5 \mu\text{m}$. Since the spherical confinement mimicking the nucleus in our model adopts a radius of 19.7σ , we have $\sigma = 0.254 \mu\text{m}$ for the reduced length unit. We further determined the reduced time unit with the expression $\tau_B = \frac{30\pi\eta\sigma^3}{k_B T}$, which was obtained by matching the diffusion coefficient from simulations with that in the nucleus (see Supplementary Material). Using a value of the nucleoplasmic viscosity as $\eta = 10^{-2} \text{ Pa}\cdot\text{s}$, we estimated $\tau_B = 3.6 \text{ s}$.

We note that the energy unit ($k_B T$) in our model should be viewed as an effective temperature instead of strictly the biological value (300 K). Since the nucleus is a non-equilibrium system with constant perturbations from molecular motors and chemical reactions, the ensemble of genome organizations collected over a population of cells as in Hi-C experiments is unlikely to be in thermodynamic equilibrium. However, non-equilibrium distributions can often be well approximated by renormalized Boltzmann distributions with effective potentials and temperatures [cite]. It is these effective quantities that we inferred from Hi-C contact maps to describe the interactions among chromosomes.

Comment 2: Related to this SI section, I found the discussion of units and orders of magnitude rather weak in the main text. Many length scales are given without units; the size of the constituents cannot be found in the main text; as for time scale one has to dig deep into the SI to realize that they have actually been given serious consideration. Similarly, the level of coarse-graining is insufficiently discussed: as I understand, individual beads on the chromatin model represent 1Mb, i.e. huge molecular weight... but it is unclear in the main text whether the "protein" beads represent one or many proteins. It would be good to discuss more precisely what these proteins are and how many of them are represented in a single MD particle (given the size of $0.12\mu\text{m}$ indicated in the SI, it is a large number).

Response:

We thank the reviewer for this comment. As mentioned in the *Response to Comment 1*, we added more text in the *Methods Section* to detail the mapping of reduced model units to physical units for length and time. In addition, we changed all units in the main text to physical units.

In the newly added sub-section of Methods, we added the following text to discuss the size of coarse-grained particles.

We note that the above estimation is crude since many additional proteins and RNA molecules are present inside the nucleoli. Therefore, the nucleolar particles should be viewed as molecular aggregates with size $\sigma_p = 0.5\sigma \sim 0.12 \mu\text{m}$, rather than a single protein molecule. Given the size of a typical protein as $5 \sim 10 \text{ nm}$ [cite], the number of molecules within a single coarse-grained particle can be on the order of 10^3 . This number, while large, is, in fact, on the same order as the amount of distinct molecules that make up the nucleoli [cite].

Given the large size of these coarse-grained particles, we have switched their name from “protein particles” to “nucleolar particles” to acknowledge their aggregate nature.

The formation of nucleoli is potentially more complicated to be captured by the coarse-grained particles used in our model. However, we carried out additional simulations to demonstrate that the stability of the multi-droplet state is robust with respect to nucleolar particle size (Fig. S18), nucleolar particle-chromatin interactions (Fig. S1), chromatin-chromatin interactions (Figs. S5, S6 and S17), and genome model resolution (Fig. S7). We anticipate the mechanism uncovered in our study to be of biological relevance.

We added the following text in the *Discussion Section* of the main text to point out the limitations and significance of our study.

Despite our best effort, the coarsened representations for the genome and nucleoli are bound to be approximate, and certain features of the biological system would inevitably be missed. While it may struggle at quantitative predictions, the model serves as a valuable tool for mechanistic explorations. In particular, since the nucleation and arrest mechanism arises from the generic polymer nature of the chromatin network, it should be insensitive to many of the model assumptions. Indeed, as shown in Fig. S17, the multi-droplet state was also observed using a genome model parameterized with Hi-C data generated from clinical tissue samples with colorectal cancer [cite]. The tumor Hi-C data and genome organization differ significantly from the one presented in the main text. The stability of the multi-droplet state is also preserved in additional perturbations that directly vary the resolution of the genome model (Fig. S7), chromosome-chromosome interactions (Figs. S5-S6), nucleolar particle size (Fig. S18), and nucleolar particle-chromatin interactions (Fig. S1).

Comment 3: *Still on the subject of dimensions and orders of magnitude, I was surprised to see energy scales of only kT involved in interactions at the Mb scale. One has to go to the SI to find mention that these are free energies with major interplay between entropy and energy. This deserves a serious discussion in the main text. Similarly, the mention of activity is only in the last sentences of the discussion. Should we really be thinking of kT as being the thermal energy*

here? Or is it an effective temperature including the nonequilibrium effects? How important are these? This deserves a more thorough discussion, and to be addressed earlier.

Response:

We thank the reviewer for this excellent comment. We note that the energy scale of our model is mostly dictated by the interaction strength within and among chromosomes. Importantly, we adjusted the strength such that statistical averages evaluated using the ensemble of genome structures following the Boltzmann distribution, $e^{-U_{\text{genome}}(\mathbf{x})/k_B T}$, reproduce Hi-C contact frequencies. Since Hi-C experiments detected genomic contacts inside the nucleus, $e^{-U_{\text{genome}}(\mathbf{x})/k_B T}$ provides an effective distribution of genome structures in situ. $k_B T$ should indeed be viewed as an effective temperature here since, as the reviewer suggested, the nucleus is a non-equilibrium system.

We added the following sentence when introducing the genome model.

Because of the non-equilibrium nature of the system, these experimentally derived interactions and temperature represent effective approximations to the steady-state distribution of genome organization (see Methods).

As mentioned in the *Response to comment 1*, more discussion on the non-equilibrium nature of the model interactions and temperature was included in the *Methods* section of the manuscript.

We added the following text in the *Methods Section* to comment on the energy scale used for nucleolar particle-chromatin interactions

The molecular aggregate interpretation of nucleolar particles is also consistent with our use of $\epsilon = 1.8 k_B T$ for specific interactions. This value appears much smaller than the strength of typical non-bonded interactions, such as hydrogen bonds. ϵ should be viewed as “free energy” that accounts for the averaging over the heterogeneous interaction pattern of different molecules and the entropic penalty arising from confining individual molecules together. Furthermore, as stated above, the effective temperature in our model is presumably higher than 300 K, resulting in more significant values for ϵ .

We note that our results are qualitatively robust with respect to the strength of nucleolar particle-chromatin interactions. As shown in Fig. 6 of the main text, the multiple-droplet state is seen in many parameter values, so is the entropic barrier for droplet coalescence. In addition to these results, we performed simulations with different combinations of the specific and non-specific interaction strength between nucleolar particles and chromatin. For each parameter set, we followed the same protocol as in the main text to simulate phase separation. For example, twelve independent 20-million-step-long trajectories were performed, and the droplet number at the end of each simulation trajectory was recorded to compute the averages.

Figure R11: **Average number of droplets at various combinations of specific and non-specific interactions between nucleolar particles and chromatin.** 12 independent simulations were performed for each set of parameters to compute the average droplet number. The star indicates the value used in simulations presented in the main text.

The average number of droplets as a function of specific and non-specific interactions is shown in Fig. R11. We found that as soon as the strength of specific interactions becomes strong enough to promote phase separation, the multi-droplet state emerges. At the strength of $1.4 k_B T$, though droplets begin to emerge in some simulations, there are still significant simulations without phase separation. The average number of droplets in some of the setups is, therefore, less than one. Droplets appear in all simulations at $1.6 k_B T$ and larger values, and many trajectories produce multiple droplets. We decided to use $1.8 k_B T$, which leads to multiple droplets in most simulations. Furthermore, the surface tension of the resulting droplets is comparable to the experimental value as well. The non-specific interaction was chosen as $1.0 k_B T$, but as the phase diagram shows, it has little impact on the results.

Comment 4: *Regarding the discussion of the literature, there has been in the past few years a significant theoretical and experimental effort to understand the mechanical interactions between phase-separating droplets and surrounding *elastic* networks (see for instance: Cell 175, 1481 (2018); Physical Review X 8, 011028 (2018); Journal of the Mechanics and Physics of Solids 145, 104153 (2020); arXiv:2102.02787). Several of these works consider the importance of chromatin elasticity on nuclear condensate formation, in particular. How would these scenarios play out in the simulations presented in this paper? Would it be possible in these simulations to characterize the stresses induced by the droplets in the chromatin network, and the corresponding mechanical pressure this induces in the droplets? I understand that the authors are taking a different point of view here and that this might be beyond the scope of this paper, but can they at least relate their*

results and model to this body of literature?

Response:

We thank the reviewer for this comment. We agree that the alternative mechanism is highly relevant and deserves further discussion.

In contrast to our focus on the viscoelastic nature of the chromatin network and its attractive interactions with phase-separating agents, the studies mentioned by the reviewer were concerned with cross-linked gels and repulsive interactions. In the presence of permanent cross-linking, droplet-induced swelling of the polymer network can give rise to compressive stresses that arrest the coarsening dynamics completely. Therefore, the thermodynamic equilibrium becomes the monodisperse state with many droplets whose size is largely determined by the network mesh. Because of their favorable interaction in our model, there is substantial wetting of the chromatin network by nucleolar particles. Therefore, we do not anticipate the presence of significant stress on the droplets.

To directly evaluate the mechanical impact of chromatin on droplets, we computed the pressure for all the droplets formed in phase separation simulations. We used the expression of local pressure introduced by [Lion and Allen, J. Condens. Matter Phys., 2012]

$$P(\vec{r}) = \frac{1}{3\Omega} \left\langle \sum_{i=1}^N \frac{|\vec{p}_i|^2}{m_i} \Lambda_i + \sum_{i=1}^{N-1} \sum_{j>i} (\vec{f}_{ij} \cdot \vec{r}_{ij}) l_{ij} \right\rangle. \quad (1)$$

Here, Ω is the volume of the region of interest, centred on \vec{r} , Λ_i is the unity if particle i lies within the volume Ω , and zero otherwise. l_{ij} is the fraction ($0 \leq l_{ij} \leq 1$) of the line joining particles i and j that lies within Ω . \vec{r}_i and \vec{r}_j are the positions of particles i and j , $\vec{r}_{ij} = \vec{r}_i - \vec{r}_j$, and \vec{f}_{ij} denotes the force exerted on particle i by particle j . \vec{p}_i and m_i are the momentum and mass of particle i . The bracket $\langle \cdot \rangle$ represents the ensemble average over independent configurations with droplet size that are within $\pm 0.0534 \mu\text{m}$ of the presented values. For simplicity, we approximated the first term as $N_d k_B T / \Omega = \rho_d k_B T$ using the equal partition theorem $\frac{|\vec{p}_i|^2}{m_i} = m_i |\vec{v}_i|^2 = 3k_B T$, where N_d and ρ_d are the number of particles and density of droplets.

As shown in Fig. R12, the pressure does not increase as droplets grow in size. This contrasts with the results shown in [Zhang et al. PRL 2021], where the pressure increases as a function of the droplet size. The distinct pressure behaviors suggest that a different mechanism from mechanical stress causes the stability of the multi-droplet state in our study.

To ensure that we computed the local pressure correctly, we performed additional simulations in which nucleolar particles and chromatin experience repulsive interactions. The same LJ potential defined in Eq. 1 of the main text was used to model these interactions, and we shifted the cutoff r_c for nucleolar particle-chromatin interactions to $1.12\sigma_l$. Nucleolar particles are still attracted to each other using the same potential as in the main text. Phase separation simulations with repulsive simulations always led to a single droplet.

Figure R12: **Pressure as a function of droplet size for the model presented in the main text (A) and for a model with repulsive interactions between nucleolar particles and chromatin (B).**

Following [Zhang et al. PRL 2021], we further introduced crosslinks between 12,000 randomly and non-repeatedly selected genomic loci pairs. Crosslinks were modeled with a harmonic potential of the form:

$$U_{\text{harmonic}}(r) = K_2(r - r_0)^2 + K_3(r - r_0)^3 + K_4(r - r_0)^4 \quad (2)$$

where $r_0 = 2.0\sigma$, $K_2 = K_3 = K_4 = 20\epsilon$. We carried out 12 independent 20-million-step-long simulations using the new setup in the energy function. These simulations were initialized from a configuration produced by the model presented in the main text. As shown in Fig. R13, crosslinking caused significant perturbations to the genome organization, with a loss of chromosome territories and the phase separation of compartments. The cross-linked network was able to stabilize the two droplets. Consistent with the result presented in Ref. [Zhang et al. PRL 2021], for this model, the pressure computed using Eq. 1 indeed increases as the droplets grow (Fig. R12B).

We added the following text in the *Discussion Section* of the manuscript to comment on the two distinct mechanisms.

The mechanism uncovered here differs from the suppression of droplet growth in an elastic matrix discussed in several recent studies [cite]. In contrast to our focus on the viscoelastic nature of the chromatin network and its attractive interactions with phase-separating agents, these studies were concerned with cross-linked gels and repulsive interactions. In the presence of permanent cross-linking, droplet-induced swelling of the polymer network can give rise to compressive stresses that arrest the coarsening dynamics completely. Therefore, the thermodynamic equilibrium becomes the monodisperse state with many droplets whose size is largely determined by the network mesh. Because of their favorable interactions in our model, there is substantial

Figure R13: Crosslinking causes significant perturbations to the genome organization. (A) An equilibrated, representative configuration of the genome obtained from simulations with random crosslinking. (B) Radial distributions of A/B compartments. An example genome configuration is shown as the inset, with the two compartments colored in red (A) and blue (B) respectively.

wetting of the chromatin network by nucleolar particles. The droplets incur minimal stress on the network (Fig. S20), and the multi-droplet state remains metastable (Fig. 3). We note that two mechanisms are not necessarily mutually exclusive and could contribute to the formation of different condensates inside the nucleus since the dynamical properties of chromatin are known to be timescale dependent [cite].

Comment 5: Fig 2E: what is the initial configuration in the simulations? Is this distribution of radial positions equilibrium, or does it just reflect the initial conditions?

Response:

The initial configuration for the genome was taken from the end configuration of a 20-million-step long simulation performed in our previous study [Qi et al. Biophys. J. 2020]. We further introduced nucleolar particles with random positions inside the spherical confinement. The radial chromosome positions shown in Fig. 2E were determined by averaging over configurations collected from the 12 independent simulations carried out in this study.

We added the following text to the *Methods Section* of the main text to detail the construction of the initial configuration

The initial configuration of MD simulations was built as follows. We first obtained chromosome conformations from the end configuration of a 20-million-step long simulation of the genome-only model carried out in our previous study [cite]. Next, 500

nucleolar particles were placed with random positions inside the spherical confinement. We further relaxed the system to avoid steric clashes by performing 400,000 step MD simulations. ϵ in Eq. 1 was set to $1.0 k_B T$ for both specific and non-specific interactions to prevent cluster formation during the equilibration period. The last configuration of this equilibration simulation was then used to initialize the phase separation simulations.

We revised the caption to clarify the data used to generate Fig 2E.

Simulated radial chromosome positions correlate strongly with experimental values [cite]. Error bars correspond to the standard deviation of the 12 mean values estimated using individual simulation trajectories.

Comment 6: *p6, the authors mention heterochromatin condensation. The current understanding of these domains is that they are intimately connected to the condensation of a protein, HP1 α , into chromatin-attracting phase-separating droplets (Nature 547, 241 (2017)). Is this effect included in the model? How?*

Response:

We note that in our model, chromosomes were represented as block co-polymers with three monomer types as A, B , and C . The chromosome-specific compartment profiles were derived from Hi-C data, and A and B compartments largely correspond to euchromatin and heterochromatin. Type C represents centromeric regions. We did not explicitly consider HP1 α or other molecules that might form droplets and compact chromatin.

However, the interactions among B compartments were parameterized from Hi-C data and should, in principle, capture the impact of these molecules effectively. For example, as shown in Fig. R14, our model indeed predicts heterochromatin to be more condensed than euchromatin of the same length. The condensation level was measured using the radius of gyration of chromosome segments consisting of only B (heterochromatin) or A (euchromatin) compartments.

We added the following sentence in the main text to highlight our use of effective interactions to capture the role of proteins and histone modifications in genome organization.

While the model does not explicitly include histone modifications, transcription factors, or molecular motors, reproducing the contact probabilities between genomic segments measured in situ effectively allows it to account for the contribution of these factors to genome organization.

Comment 7: *The quantitative definition of R_g should come much, much earlier, given how central this is... I got pretty annoyed trying to guess it.*

Figure R14: **Optimized genome model predicts more expanded configurations for euchromatin than heterochromatin.** Euchromatin (heterochromatin) were defined as regions with a continuous stretch of A (B) compartments. The errorbars measure the standard deviation of the radius of gyration for domains that are of the same genomic length.

Response:

We thank the reviewer for this comment. We have moved the definition of Rg from the Method section to the place that it was first introduced in the main text. The updated text now reads

To better understand the stability of the multi-droplet state, we computed the free energy profile as a function of the radius of gyration (R_g) for a two-droplet system (see Methods). R_g is defined as $\sqrt{\frac{1}{N} \sum_{i=1}^N |\mathbf{r}_i - \mathbf{r}_{com}|^2}$, where \mathbf{r}_i is the coordinate of the i th nucleolar particle and \mathbf{r}_{com} corresponds to the center of mass. The summation includes all N nucleolar particles in either one of the two droplets. As the size of individual droplets remains stable, ...

Comment 8: *p15, $\sigma = 0.5, 0.75$ and 0.75 for protein-protein, protein-NAD and protein-nonNAD interactions." Why these values? How did the authors determine them?*

Response:

As noted in the section *Estimating the Size and Number of Nucleolar Particles*, we determined the size of nucleolar particles based on the overall size of nucleoli to 0.5σ , where σ is the size of chromatin beads. Since σ_l in the Lennard-Jones potential represents the distance at which excluded volume effect becomes apparent, we chose its value to be $\sigma_p = 0.5\sigma$ for the interaction between nucleolar particles. For the interactions between protein and

chromatin, we chose σ_l as the arithmetic average between σ_p and σ , which leads to 0.75σ . Such a combinatorial procedure is frequently used in deriving atomistic force fields.

We added the following text to the manuscript to clarify our parameter choice.

The values for σ_l were chosen based on the size of nucleolar particles ($\sigma_p = 0.5\sigma$) and chromatin beads (σ) with arithmetic averaging.

RESPONSE TO REFEREE 3

Comment 0: *This manuscript titled “In this work by Qi and Zhang, authors present a computational study that investigates coarsening of liquid droplets in the cell nucleus in the presence of chromatin polymer network. Their computational model specifically focuses on exploring the role of chromatin in the thermodynamics and kinetics of the phase separation, by which these droplets form. Authors suggest that a two-droplet state is a metastable state and separated from a one-droplet state by an entropic barrier. They also propose the protein-chromatin interactions to facilitate the droplet nucleation, but to hinder the droplet coarsening due the correlated motion between droplets. The main message of the paper is that protein-chromatin interactions arrest the phase separation of droplets in a multi-droplet state.*

This paper asks an interesting question about the role of chromatin network in the phase separation of nuclear liquid droplets and nicely reproduces previous experimental observations. However, it is not immediately clear what new physical or biological insights it reveals. It appears that authors have mainly focused on reproducing the experimentally observed phenomenology, but have missed out on an opportunity to report on the behavior of the chromatin network, which in opinion of this reviewer, is the novel part in this work. Yet, this part of the manuscript is presently rather underdeveloped. Therefore, I cannot recommend publication of this paper in Nature Communications in its present form.

Response:

We appreciate the reviewer’s assessment of the paper, and we thank him/her for the detailed suggestions and comments.

We want to point out that, as the reviewer stated, phase separation of nuclear liquid droplets is an important topic that has received a lot of attention recently. In particular, several aspects of the phase separation appear to defy predictions of the classical nucleation theory, and it is not clear what drives the emergence and stability of multiple droplets. Resolving the mystery around the multi-droplet state could improve understanding of biological condensate formation and phase separation in general.

To the best of our knowledge, the nucleation and arrest mechanism uncovered in our study has not been discussed in the existing literature. It highlights the importance of attractive interactions between nuclear bodies and chromatin on the thermodynamics and kinetics of phase separation. The fact that our model can reproduce several previous experimental observations lends support to the biological relevance of this mechanism. We believe that discovering a new mechanism is significant as it provides a fresh perspective on nuclear phase separation.

We agree that our effort in building a biologically significant genome organization from Hi-C data differentiates the current work from other studies that consider generic polymer networks. The algorithm and energy function used for modeling the chromatin network was introduced in our previous work [Qi et al. Biophy J, 2020]. Following the reviewer’s suggestion, we have provided more details on the model energy function in the Supplementary Material (see *Response to Comment 2*).

We would like to emphasize that all results presented in the main text were obtained using simulations with the presence of the chromatin network, including the free energy profiles in Figs 3 and 6 and the abnormal diffusion in Fig 5. These results contrast with the ones obtained using simulations performed without the chromatin network, i.e., when the chromatin was dissolved into individual particles (Fig S3 and S14). We have revised the main text in numerous places to point out the contribution of the chromatin network to the entropic barrier and to the slow dynamics.

Interestingly, while the chromatin network is crucial for stabilizing the multi-droplet state, the exact interactions that make up the network is less important. As detailed in *Response to Comment 4*, large perturbations to chromosome-chromosome interactions do not vary the stability of the multi-droplet state significantly, supporting the robustness of the uncovered mechanism with respect to the chromatin network. We introduced new text in the *Discussion Section* of the main text to discuss the sensitivity of our finding with respect to details in modeling the chromatin network. The relevant text is quoted below.

Despite our best effort, the coarsened representations for the genome and nucleoli are bound to be approximate, and certain features of the biological system would inevitably be missed. While it may struggle at quantitative predictions, the model serves as a valuable tool for mechanistic explorations. In particular, since the nucleation and arrest mechanism arises from the generic polymer nature of the chromatin network, it should be insensitive to many of the model assumptions. Indeed, as shown in Fig. S17, the multi-droplet state was also observed using a genome model parameterized with Hi-C data generated from clinical tissue samples with colorectal cancer [cite]. The tumor Hi-C data and genome organization differ significantly from the one presented in the main text. The stability of the multi-droplet state is also preserved in additional perturbations that directly vary the resolution of the genome model (Fig. S7), chromosome-chromosome interactions (Figs. S5-S6), nucleolar particle size (Fig. S18), and nucleolar particle-chromatin interactions (Fig. S19).

The role of protein-chromatin interactions in phase separation and stabilizing the multi-droplet state differs significantly from another mechanism, which arises from the pressure of the polymer network [Zhang et al. PRL 2021]. In this alternative mechanism, protein-chromatin interactions are not important. In fact, it is the opposite, as the two are expected to have repulsive interactions. In our opinion, the repulsive mechanism is less robust due to its sensitivity to polymer network mesh size, nucleolar particle size, etc. On the contrary, the stability of the multi-droplet state in our model does not depend on these factors (see *Response to Comment 4*). We added the following text in the *Discussion Section* to directly compare and contrast the two mechanisms

The mechanism uncovered here differs from the suppression of droplet growth in an elastic matrix discussed in several recent studies [cite]. In contrast to our focus on the viscoelastic nature of the chromatin network and its attractive interactions with phase-separating agents, these studies were concerned with cross-linked gels and repulsive interactions. In the presence of permanent cross-linking, droplet-induced swelling

of the polymer network can give rise to compressive stresses that arrest the coarsening dynamics completely. Therefore, the thermodynamic equilibrium becomes the monodisperse state with many droplets whose size is largely determined by the network mesh. Because of their favorable interactions in our model, there is substantial wetting of the chromatin network by nucleolar particles. The droplets incur minimal stress on the network (Fig. S20), and the multi-droplet state remains metastable (Fig. 3). We note that two mechanisms are not necessarily mutually exclusive and could contribute to the formation of different condensates inside the nucleus since the dynamical properties of chromatin are known to be timescale dependent [cite].

Comment 1: *The Introduction is weakly referenced, with only 26 references out of 90 across manuscript and supplements, and with seminal experimental observations, which should be prominent in the Introduction, only appearing in the later parts of the paper or not at all.*

Response:

We revised the introduction and added a collection of citations, especially the ones on prior experimental studies. It now includes 49 references.

Comment 2: *The title of the paper focuses on the role of chromatin in coalescence of nuclear droplets, yet the manuscript only briefly describes the genome. The description of the genome part of the model is quite minimalistic, mostly directing the reader to previous publications. For example, the interaction potentials are only listed, but not explicitly given, similar for the initial polymer configurations that were obtained from HiC data. These should be all explicitly listed/described in the supplements of the present paper.*

Response:

We thank the reviewer for this helpful comment. We added detailed descriptions of the energy function for chromosomes in the Supplementary Material. The updated text now reads

The energy function of the genome model is defined as

$$U_{\text{Genome}}(\mathbf{r}) = U(\mathbf{r}) + U_{\text{intra}}(\mathbf{r}) + U_{\text{inter}}(\mathbf{r}) + U_{\text{Xi}}(\mathbf{r}) + U_{\text{inter}}^{\text{specific}}(\mathbf{r}). \quad (3)$$

$U(\mathbf{r})$ represents a generic potential applied to each chromosome to ensure the polymeric topology of chromosomes:

$$U(\mathbf{r}) = \sum_i [u_{\text{bond}}(r_{i,i+1}) + u_{\text{angle}}(\vec{r}_{i,i+1}, \vec{r}_{i+1,i+2}) + u_c(r_i)] + \sum_{j>i} u_{\text{sc}}(r_{ij}), \quad (4)$$

where $u_{\text{bond}}(r_{i,i+1})$ and $u_{\text{angle}}(r_{i,i+1}, r_{i+1,i+2})$ are the bonding and angular potential applied for neighboring beads to ensure the connectivity of the chromatin chain.

$$u_{\text{bond}}(r_{i,i+1}) = -\frac{1}{2}KR_0^2 \ln \left[1 - \left(\frac{r_{i,i+1}}{R_0} \right)^2 \right], K_b = 30\epsilon, R_0 = 1.5\sigma$$

$$u_{\text{angle}}(\vec{r}_{i,i+1}, \vec{r}_{i+1,i+2}) = K_a [1 - \cos(\theta - \pi)], K_a = 2\epsilon, \cos\theta = \frac{\vec{r}_{i,i+1} \cdot \vec{r}_{i+1,i+2}}{|\vec{r}_{i,i+1}| \cdot |\vec{r}_{i+1,i+2}|} \quad (5)$$

$u_c(r_i)$ is a spherical boundary potential applied to each bead to mimic the confinement effect of nuclear envelop.

$$u_c(r_i) = \begin{cases} U_{\text{LJ}}(r_i), & r_i \leq 2^{1/6}\sigma \\ 0, & r_i > 2^{1/6}\sigma \end{cases} \quad (6)$$

where $U_{\text{LJ}}(r) = 4\epsilon \left[\left(\frac{\sigma}{r} \right)^{12} - \left(\frac{\sigma}{r} \right)^6 \right]$ is the Lennard-Jones potential. r_i is the distance between i -th bead and the wall surface. $u_{\text{sc}}(r_{ij})$ is a non-bonded soft-core potential added to each pair formed by beads index i and j to account for the excluded volume effect while allowing finite probability of cross-over of polymer chains.

$$u_{\text{sc}}(r_i) = \begin{cases} 0.5E_{\text{cut}} \left(1 + \tanh \left[\frac{2U_{\text{LJ}}(r_i)}{E_{\text{cut}}} - 1 \right] \right), & r_i \leq r_{\text{cut}} \\ U_{\text{LJ}}(r_i), & r_{\text{cut}} < r_i \leq 2^{1/6}\sigma \\ 0, & r_i > 2^{1/6}\sigma \end{cases} \quad (7)$$

which corresponds to the Lennard-Jones potential capped off at a finite volume within a repulsive core to allow for chain crossing at finite energy cost. $E_{\text{cut}} = 4\epsilon$ and r_{cut} is chosen as the distance at which $U_{\text{LJ}}(r) = 0.5E_{\text{cut}}$.

$U_{\text{intra}}(\mathbf{r})$ is the intra-chromosomal potential applied to genomic loci within the same chromosome, while $U_{\text{inter}}(\mathbf{r})$ is similarly defined but for interactions between loci from different chromosomes. $U_{X_i}(\mathbf{r})$ is a weakly attractive potential applied to only one of the two X chromosomes to induce the known X-chromosome inactivation. These terms adopt the following form:

$$U_{\text{intra}}(\mathbf{r}) = \sum_I \sum_{i,j \in I} \left[\alpha_{\text{ideal}}(|i-j|) + \alpha_{\text{intra}}(T_i^I, T_j^I) \right] f(r_{ij}) \quad (8)$$

where I indexes over each chromosome and i and j index over pair of beads on that chromosome. $\alpha_{\text{ideal}}(|i-j|)$ is a function depends only on the sequence separation between two beads i and j . $\alpha_{\text{intra}}(T_i^I, T_j^I)$ depends specifically on the compartment types T_i^I and T_j^I , which can be A, B or C. $f(r_{ij})$ measures the probability of contact formation for two loci separated by a distance of r_{ij} , and its ensemble average corresponds to the contact probability measured in Hi-C experiments. $f(r_{ij})$ adopts the form:

$$f(r_{ij}) = \begin{cases} \frac{1}{2} [1 + \tanh [\eta(r_c - r_{ij})]], & r_{ij} \leq r_c \\ \frac{1}{2} (r_c/r)^4, & r_{ij} > r_c \end{cases} \quad (9)$$

where $r_c = 1.5$ and $\eta = 2.5$. Similarly, we have

$$U_{\text{inter}}(\mathbf{r}) = \sum_{I,J} \sum_{i \in I, j \in J} \alpha_{\text{inter}}(T_i^I, T_j^J) f(r_{ij}), \quad (10)$$

and

$$U_{X_i}(\mathbf{r}) = \sum_{i,j \in X_i} w(r_{ij}) = \sum_{i,j \in X_i} \alpha_{X_i}(|i-j|) f(r_{ij}) \quad (11)$$

The last term in the energy function, $U_{\text{inter}}^{\text{specific}}(\mathbf{r})$, characterizes the specific compartmental inter-chromosome interaction beyond the generic compartmentalization potential, and adopts the form

$$U_{\text{inter}}^{\text{specific}}(\mathbf{r}) = \sum_{I,J} \sum_{i \in I, j \in J} \alpha_{\text{inter}}^{IJ}(T_i^I, T_j^J) f(r_{ij}) \quad (12)$$

Practically, this term was combined with the third term $U_{\text{inter}}(\mathbf{r})$ to describe the inter-chromosome interactions using one set of parameters.

Mathematical expressions for the various energy terms in $U_{\text{Genome}}(\mathbf{r})$ were designed such that their ensemble averages can be mapped onto combinations of contact frequencies measured in Hi-C. The correspondence between the energy functions and Hi-C measurements allows model parameterization with an efficient maximum entropy optimization algorithm. Specifically, $\alpha_{\text{ideal}}(|i-j|)$, $\alpha_{\text{intra}}(T_i^I, T_j^J)$, $\alpha_{\text{inter}}(T_i^I, T_j^J)$, and $\alpha_{X_i}(|i-j|)$ were tuned to satisfy the following constraints:

$$\begin{aligned} \left\langle \sum_{I \neq X} \sum_{i,j \in I} f(r_{ij}) \delta_{|i-j|,s} \right\rangle_{U_{\text{Genome}}(\mathbf{r})} &= \sum_{I \neq X} \sum_{i,j \in I} f_{ij}^{\text{exp}} \delta_{|i-j|,s}, \quad \text{for } s = 1, \dots, n-1 \\ \left\langle \sum_I \sum_{i,j \in I} f(r_{ij}) \delta_{T_i^I, T_1} \delta_{T_j^I, T_2} \right\rangle_{U_{\text{Genome}}(\mathbf{r})} &= \sum_I \sum_{i,j \in I} f_{ij}^{\text{exp}} \delta_{T_i^I, T_1} \delta_{T_j^I, T_2}, \quad \text{for } T_1, T_2 \in \{A, B, C\} \\ \left\langle \sum_{I,J} \sum_{i \in I, j \in J} f(r_{ij}) \delta_{T_i^I, T_1} \delta_{T_j^J, T_2} \right\rangle_{U_{\text{Genome}}(\mathbf{r})} &= \sum_{I,J} \sum_{i \in I, j \in J} f_{ij}^{\text{exp}} \delta_{T_i^I, T_1} \delta_{T_j^J, T_2}, \quad \text{for } T_1, T_2 \in \{A, B, C\} \\ \left\langle \sum_{i,j \in X} f(r_{ij}) \delta_{|i-j|,s} \right\rangle_{U_{\text{Genome}}(\mathbf{r})} &= \sum_{i,j \in X} f_{ij}^{\text{exp}} \delta_{|i-j|,s}, \quad \text{for } s = 1, \dots, n_X - 1 \\ \left\langle \sum_{i \in I, j \in J} f(r_{ij}) \delta_{T_i^I, T_1} \delta_{T_j^J, T_2} \right\rangle_{U_{\text{Genome}}(\mathbf{r})} &= \sum_{i \in I, j \in J} f_{ij}^{\text{exp}} \delta_{T_i^I, T_1} \delta_{T_j^J, T_2}, \quad \text{for } T_1, T_2 \in \{A, B\} \end{aligned} \quad (13)$$

where $\delta_{T_i^I, T_1}$ is the Kronecker delta function with the following definition:

$$\delta_{T_i^I, T_1} = \begin{cases} 1, & \text{if } T_i^I = T_1 \\ 0, & \text{otherwise} \end{cases} \quad (14)$$

The angular bracket represents the ensemble average over the Boltzmann distribution $e^{-\beta U_{\text{Genome}}(\mathbf{r})}$ and f_{ij}^{exp} is the corresponding experimental contact frequency.

We applied an iterative algorithm to derive the values for $\alpha_{\text{ideal}}(|i-j|)$, $\alpha_{\text{intra}}(T_i^I, T_j^I)$, $\alpha_{\text{inter}}(T_i^I, T_j^I)$, $\alpha_{X_i}(|i-j|)$ and $\alpha_{\text{inter}}^{IJ}(T_i^I, T_j^J)$ that enforce the constraints defined in Eq. 13. As shown in [cite], while the model was parameterized only with population Hi-C data, it succeeded in reproducing a variety of observations from imaging studies. For example, A/B compartments were shown to occupy distinct nuclear regions, with B compartments preferentially at the periphery. The model further captures the formation of chromosome territories, the clustering of centromeric regions, and the radial position of individual chromosomes.

The initial chromosome conformations were obtained from the end configuration of a 20-million-step long simulation of the genome-only model carried out in our previous study. We added the following text to the *Methods Section* of the main text to provide more details on building the initial configuration

The initial configuration of MD simulations was built as follows. We first obtained chromosome conformations from the end configuration of a 20-million-step long simulation of the genome-only model carried out in our previous study [cite]. Next, 500 nucleolar particles were placed with random positions inside the spherical confinement. We further relaxed the system to avoid steric clashes by performing 400,000 step MD simulations. ϵ in Eq. 1 was set to 1.0 $k_B T$ for both specific and non-specific interactions to prevent cluster formation during the equilibration period. The last configuration of this equilibration simulation was then used to initialize the phase separation simulations.

We also provided the source code and all simulation files of the model in the github repository: <https://github.com/ZhangGroup-MITChemistry/DropletCoalescence/> to ensure reproducibility.

We want to emphasize that the results presented in the main text are robust with respect to the initial conformations of chromosomes. The 12 simulations presented in Fig. 2B of the main text were initialized with the same configuration but different random velocities. We carried out 12 additional 20-million-step-long simulations to further evaluate the robustness of our results with respect to the initial configuration. These new simulations were initialized with different configurations prepared as following. We first collected 12 uncorrelated sets of chromosome conformations from a long simulation trajectory of the genome-only model at equal time intervals. For each set of conformations, we then introduced nucleolar particles and relaxed the resulting structures following the same procedure detailed in the *Methods Section* of the main text. As shown in Fig. R15, the distribution for the number of droplets recorded at the end of these simulations is quantitatively comparable to that shown in Fig 2B.

Figure R15: **Probability distribution of the number of droplets observed at the end of 12 additional simulation trajectories initialized with different starting configurations.**

Comment 3: *Authors do not describe how was the polymeric model equilibrated and how was the equilibrium defined, which can critically impact the observed entropic barrier between the multi-droplet state and one-droplet state. The polymer dynamics in different stages of the equilibration can impact the observed droplet formation and thus change the observed outcome of the model. This should be explained in detail and controls should be provided.*

Response:

We thank the reviewer for this insightful comment. We note that our model represents the genome at a 1MB resolution. While this coarse-grained representation introduces approximations, and fine-scale features of the genome, including loops and TADs, cannot be captured, it does have significant advantages. One of them is that we can indeed carry out long-timescale simulations to equilibrate the system truly. Polymers are relatively short in contour length. They do not exhibit reptation dynamics or topological entanglements that can dramatically slow down the relaxation kinetics.

Several lines of evidence support the equilibration of our simulations

- To quantify the relaxation timescales in the system, we computed the auto-correlation function for R_g and pair-wise inter-chromosome distances d_{ij} . R_g measures the conformational relaxation of individual chromosomes, while d_{ij} quantifies the dynamics of global genome organization. As shown in Fig. R16, the correlation functions for both variables decay to zero on the order of half a million timesteps, a value that is much smaller than the total number of simulations steps (20 million) used in the main text.

We added the following text in the manuscript to highlight the simulation length.

These simulations lasted for 20 million steps, much longer than the relaxation timescale of chromosome conformations (Fig. S2).

Figure R16: The correlation time of representative collective variables is much shorter than the length of simulated trajectories, supporting their equilibration. Autocorrelation function of the radius of gyration of chromosome 11 (A) and 17 (B) as a function of time computed using simulations performed with the model presented in the main text (WT), with the model of 100kb resolution (100kb), with the model that removes inter-chromosomal interactions (no inter-), and with the model that removes both intra- and inter-chromosomal interactions (no intra-/inter-). (C) Autocorrelation function of the center of mass distance between chromosome 11 and 17 as a function of time computed using the same simulation data in A and B. Numbers in the parentheses are values of the characteristic decay timescale (τ) fitted using the function $\exp(-t/\tau)$.

- As mentioned in the *Response to Comment 2*, the results presented in the main text are robust with respect to the initial chromosome conformations. This robustness provides strong evidence that the simulation length is sufficient for relaxing the system to derive true equilibrium results.
- The error bars for the free energy calculations also support the convergence of our simulations. As detailed in the newly added text of the *Methods Section* and quoted below, these error bars were estimated via block averaging.

To compute the error bars and evaluate simulation convergence, we divided the simulation trajectories into five consecutive blocks with equal length. Free energy profiles were then calculated using only data collected from each block with the weighted histogram method (WHAM), and error bars were determined as the standard deviation of the five independent estimations.

As shown in Fig. 3 of the main text, the error bars are on the order of $\sim 0.5 k_B T$ and are much smaller than the barrier height ($\sim 7 k_B T$), supporting the significance of the entropic barrier.

We note that for the free energy calculations, in addition to long-timescale simulations, we further introduced temperature replica-exchange to facilitate convergence and equilibration. Replica-exchange molecular dynamics simulations are known to

preserve the Boltzmann distribution but vastly accelerate convergence since high-temperature simulations can help barrier crossing.

We added the following text in the manuscript to highlight the significance of the entropic barrier

The barrier height is much larger than the error bars ($0.5 k_B T$) estimated via block averaging (see Methods), supporting its statistical significance.

- As additional controls, we carried out simulations of systems that remove inter-chromosomal interactions, systems that remove both inter and intra-chromosomal interactions, and systems parameterized with tumor Hi-C data. Because of the weakened interactions, the relaxation timescales in these simulations are expected to be faster than the one presented in the main text. In all cases, we observed the multi-droplet state (see *Response to Comment 4*), supporting the presence of the entropic barrier in these systems.
- We also carried out simulations with a 100kb resolution model for the genome. Due to the ten times higher resolution than the model presented in the main text, the polymers are much longer in the absolute number of beads. Therefore, the relaxation timescales in terms of simulation time steps also increase significantly. However, this model again supports the stability of multi-droplet state, and correspondingly, the presence of an entropic barrier (see *Response to Comment 4*).

We added the following text in the manuscript to emphasize the robustness of simulation results

Notably, the emergence of the multi-droplet state is insensitive to the configuration used to initialize the simulations (Fig. S4), the interactions between chromosomes (Figs. S5-S6), and the resolution of the genome model (Fig. S7).

Comment 4: *Authors mention the viscoelastic nature of the genome playing a key role in the droplet coarsening, yet a physical explanation of this point is presently missing. The model explicitly accounts for specific interactions between proteins, and proteins with chromatin, and their respective excluded volume in form of the Lenard-Jones potential. However, the choice of the size of particles will change these interactions, altering the effect of the polymer mesh size on the protein particles and hence the timescale of their phase separation and droplet formation. Similarly, changes in the polymer part of the model, for example, the bead size will effectively change the polymer meshsize. In addition, the interactions chosen for the monomers in the genome will impact polymer relaxation times, which will impact droplet formation. Could authors elaborate on this? How were the parameters chosen? Was a phase behavior of different values of interaction parameters, proteins sizes, etc explored and relevant timescales obtained?*

Response:

We thank the reviewer for this insightful comment. The role of the genome in droplet coarsening is most evident from its contribution to the entropic barrier. We have revised the relevant text to clarify this point, and the updated text is quoted below.

The thermodynamic analysis presented in Fig. 3 suggests that the chromatin network acts much like entropic springs to hinder the coalescence of droplets [cite]. As the droplets move close to each other, they pull on the network and restricts the accessible polymer configurations. Restoring forces from the network to maximize configuration entropy counters the merging and gives rise to the barrier.

The most direct evidence for the contribution of chromatin network to the entropic barrier is shown in Fig. S3. In these independent sets of simulations, the connectivity between genomic segments is removed. For example, chromatin is dissolved from connected polymers to separate particles. Dynamical simulations of the dissolved network resulted in mono-droplet states. Furthermore, free energy calculations showed that the entropic barrier disappears in this system as well. Since the only perturbation to the system is removing the chromatin network, these results support its essential role, and the elastic nature of the network, in giving rise to the barrier. The viscoelastic role of the network is evident in the slow down of cluster diffusion shown in Fig. S16 as well.

Parameters in the model were calibrated to mimic the biological system as accurately as possible. For example, as noted in the main text, interactions among chromatin beads were parameterized to reproduce the corresponding experimental Hi-C contact frequencies using the maximum entropy optimization algorithm. We and others have shown that the resulting models produce realistic structures consistent with imaging studies for interphase [Zhang and Wolynes, PNAS, 2015] and metaphase chromosomes [Zhang and Wolynes, PRL, 2016] and the whole genome [Qi et al. Biophys J, 2020; Tjong et al. PNAS 2016].

The size of the nucleolar particles was chosen to reproduce the concentration of the nucleolar protein NPM1 and the size of the nucleoli. Details were provided in the *Section: Estimating the Size and Number of Nucleolar Particles*, which have been moved to the main text and are quoted below.

The number of nucleolar particles used in simulations were selected based on the concentration of a representative nucleolar protein NPM1 as follows. For nucleoli of $2R_{Nu}$ in diameter, and a protein concentration of $1 \mu M$ [cite], the number of nucleolar particles can be estimated as $N_p = \frac{4\pi}{3} \cdot N_A \cdot R_{Nu}^3 \cdot 1\mu M$, where N_A is the Avogadro constant. We used 500 particles in simulations, and the corresponding $R_{Nu} = 0.58 \mu m$ matches well with the observed size of nucleoli [cite].

The size of the nucleolar particles can be determined assuming a space-filling model for the nucleoli as

$$\frac{\frac{4\pi}{3} \cdot \left(\frac{2^{\frac{1}{6}} \cdot \sigma_P}{2}\right)^3 \cdot N_P}{\frac{4\pi}{3} \cdot R_N^3} = \left(\frac{R_{Nu}}{R_N}\right)^3. \quad (15)$$

Using $N_P = 500$, $R_N = 19.7$, $\sigma = 5 \mu m$ and $R_{Nu} = 0.58 \mu m$, we have the diameter of nucleolar particles $\sigma_P = 0.5\sigma$. $2^{1/6}\sigma_P$ is the equilibrium distance between neighboring nucleolar particles in the Lennard-Jones potential.

The agreement of simulated phase separation kinetics (Fig. 5 of the main text) and coalescence kinetics (Fig. 4 of the main text) with prior experimental studies further supports the biological relevance of simulation results.

However, we acknowledge that despite our best effort in parameterizing the model, it is still an approximation to the real nucleus. Therefore, it is useful to explore the robustness of our simulation results with respect to the parameter values. To examine the dependence of simulation results on the size of nucleolar particles, we altered the value of σ_P from 0.5σ to $0.3\sigma, 0.4\sigma, 0.6\sigma, 0.7\sigma, 0.8\sigma$. All other parameters in the model were kept the same. For each value of σ_P , we carried out 12 independent simulations and calculated the number of droplets formed at the end of each simulation. As shown in Fig. R17, the multi-droplet state appears for all parameter values. It is also more populated than the mono-droplet state except for $\sigma_P = 0.8\sigma$. However, at $\sigma_P = 0.8\sigma$, the nucleoli size would be slightly out of the reasonable range of experimental values [Su et al. Cell 2020]. The size of the nucleoli calculated as twice of the radius of gyration is estimated to be $\sim 7\sigma$. Since the nucleus is typically ellipsoidal while we modeled it as a sphere for simplicity, it would be more reasonable to compare the ratio between the nucleolar and nucleus size $\frac{R_{Nu}}{R_N}$. For $\sigma_P = 0.8\sigma$, the ratio is estimated to be ~ 0.18 , while the maximum ratio between the largest nucleolar size and the longest nuclear axis estimated using microscopy images is around ~ 0.13 . Thus, we conclude our simulation results are robust with respect to variations in nucleolar particle size.

We further studied the impact of the chromatin network's mesh size on droplet formation and coarsening. First, we introduced an algorithm to probe the mesh size in simulated genome structures. For a given structure, we carried out the Voronoi tessellation to tile the 3D space using the Cartesian coordinates of all monomers [Rycroft, Chaos 2009]. This tessellation allows the assignment of a spatial region, in which all points are closer to a given monomer than to any other monomers. The volume of this region provides a quantitative metric on the free space between monomers, and correlates well with the mesh size of the network. We note that our model does not have permanent cross-links, so we cannot define the mesh size in the most strict sense.

Next, we carried out two perturbations to the whole-genome model by removing inter-chromosomal interactions or by removing both intra- and inter-chromosomal interactions. As the reviewer suggested, the mesh size depends on the interaction parameters among chromatin monomers. We anticipate these two perturbations will alter the mesh

Figure R17: The stability of the multi-droplet state is robust with respect to nucleolar particle size. 12 independent simulations were performed following the same protocol as that in the main text with new values of nucleolar particle size as indicated in the title.

size dramatically. Indeed, as shown in Fig. R18, the average volume per bead increases significantly upon perturbation. The genome organizations are altered in these perturbed models as well (Fig. R19 and Fig. R20). As mentioned in the *Response to Comment 3*, the relaxation timescales in these perturbed models differ from the one presented in the main text.

For each of the perturbed models, we carried out 12 independent simulations to determine the number of droplets using the same protocol as in the main text. Despite the changes in the average volume per bead and mesh size, in both cases, the multi-droplet state remains the dominant one compared to the mono-droplet state.

As another test on the robustness of our observations with respect to the genome structure and network mesh size, we carried out additional simulations of the phase separation in a tumor model. As shown in our previous study [Jonestone et al. Cell 2020] and Fig. R21, genome organization in tumor cells differ significantly from that in normal cells, with a dramatic rearrangement of heterochromatin from the nuclear envelope to the interior. This change can be partially understood from the differences in average contact frequencies between GM12878 and tumor cells, especially for the inter-chromosomal interactions. The average volume per monomers estimated using the tumor model, again, differs from the value for GM12878 cells used in the main text (Fig. R18).

Following the same protocol as outlined in the main text, we carried out 12 independent dynamical simulations starting from randomly distributed nucleolar particles with the presence of the tumor genome model. As shown in Fig. R21, despite our use of an entirely different genome model, the multi-droplet state remains the dominant state in these simulations. Interestingly, the average number of droplets is slightly higher than the one obtained for GM12878 cells, despite a similar setup in both cases. The appearance of more nucleoli in tumor cells is indeed well known in prior experimental studies [Farley et al. Chromosoma 2015; Derenzini et al. Acta Histochem. 2017; Weeks et al. Cell. Mol. Life Sci. 2019].

In summary, the nucleation and arrest mechanism uncovered in our study for stabilizing the multi-droplet state is robust with respect to nucleolar particle size, chromatin net-

Figure R18: **Comparison of the probability distributions of the average volume per monomer across various genome models.** Since chromatin in our model is not cross-linked, average volume per monomer provides an estimation on the space available for the phase separation of nucleolar particles.

work mesh size, and model relaxation timescale. It is robust because the entropic barrier shown in Fig. 3 of the main text arises from the generic polymer nature of the chromatin network. Only when we dissolve the polymer network do we see the disappearance of the multi-droplet state Fig S3. As mentioned in the *Response to Comment 1*, our mechanism differs from the one proposed by Wingreen and coworkers [Zhang et al. PRL 2021]. In this alternative mechanism, nucleolar particles and chromatin share repulsive interactions, and the stability of the multi-droplet state arises from compressive stresses of the chromatin network. The repulsive mechanism is, in fact, expected to be sensitive to polymer network mesh size, nucleolar particle size, etc.

Figure R19: **The genome organization obtained after removing Hi-C optimized inter-chromosome interactions, while differs significantly from the one presented in the main text, supports the stability of the multi-droplet state.** For comparison, see Fig. 2 of the main text. (A) Representative configuration of the genome that illustrates the formation of chromosome territories. (B) Correlation between chromosome radial positions obtained using the perturbed model and the original results presented in the main text. (C) Radial distributions of *A/B* compartments. An example genome configuration is shown as the inset, with the two compartments colored in red (*A*) and blue (*B*) respectively. (D) Probability distribution of the number of droplets observed at the end of 12 independent simulation trajectories.

Figure R20: **The genome organization obtained after removing Hi-C optimized intra- and inter-chromosome interactions, while differs significantly from the one presented in the main text, supports the stability of the multi-droplet state.** For comparison, see Fig. 2 of the main text. (A) Representative configuration of the genome that illustrates the formation of chromosome territories. (B) Correlation between chromosome radial positions obtained using the perturbed model and the original results presented in the main text. (C) Radial distributions of *A/B* compartments. An example genome configuration is shown as the inset, with the two compartments colored in red (*A*) and blue (*B*) respectively. (D) Probability distribution of the number of droplets observed at the end of simulation trajectories.

Figure R21: **Tumor genome organization, while differs significantly from the one presented in the main text, supports the stability of the multi-droplet state.** For comparison, see Fig. 2 of the main text. (A, B) Average intra (A) and inter (B) chromosome contact probabilities between various compartments estimated using Hi-C data for GM12878 (blue) and tumor (red) cells. (C) Representative configuration of the genome that illustrates the formation of chromosome territories. (D) Correlation between chromosome radial positions obtained using simulations of the GM12878 and tumor genome model. (E) Radial distributions of A/B compartments. An example genome configuration is shown as the inset, with the two compartments colored in red (A) and blue (B) respectively. (F) Probability distribution of the number of droplets observed at the end of simulation trajectories.

Comment 5: On p. 9 authors state: “The thermodynamic analysis suggests that the chromatin network acts much like rubber with entropic springs to hinder the coalescence of droplets.” Authors do not explain this analysis, nor do they provide reference. This should be elaborated upon and put into the context of the chromatin viscoelasticity impacting timescales and length scales of droplet coalescence.

Response:

We apologize for the lack of clarity. The “thermodynamic analysis” refers to our free energy decomposition presented in Fig. 3 of the main text. As shown in Fig. 3B-C, the entropy decreases (contributions from entropic part $-T\Delta S$ increase) when R_g decreases (droplets coalesce). Importantly, the entropic barrier mainly arises from the chromatin network, as dissolving the network removes the barrier completely (Fig. S3). It is this finding that prompted us to propose the analogy between chromatin network and rubber. As discussed in the book by Flory [Principles of Polymer Chemistry], rubbers are essentially polymer networks with entropic restoring forces.

We have revised the text to be more explicit about the argument.

The thermodynamic analysis presented in Fig. 3 suggests that the chromatin network acts much like entropic springs to hinder the coalescence of droplets [cite]. As the droplets move close to each other, they pull on the network and restrict the accessible polymer configurations. Restoring forces from the network to maximize configurational entropy counter droplet merging and give rise to the barrier. A similar mechanism could potentially impact the coarsening dynamics and the pathway leading to the formation of multiple droplets.

We expect that the chromatin network will impact the coalescence of the final two droplets and the coarsening dynamics of smaller clusters. As shown in Fig. 5 and 6, the coarsening dynamics is dominated by Brownian diffusion mechanisms. The presence of the barrier slows the growth substantially. We added the following sentences to further highlight this point and connect Fig. 3 with the coarsening dynamics.

The sub-diffusive motion arises from both the elastic stress produced by the viscoelastic chromatin network and the physical tethering of droplets to the chromosomes. In addition to the sub-diffusive motion, the chromatin network could further reduce the exponent and slow down the Brownian diffusion dominated coarsening dynamics by hindering droplet coalescence through entropic barriers similar to that shown in Fig. 3.

Comment 6: The explanation for the choice of the interaction strengths for specific and non-specific interactions is missing. Such choice requires an optimization of interactions in form of a phase diagram. It would be helpful, if authors could provide this or otherwise motivate their choice.

Figure R22: **Average number of droplets at various combinations of specific and non-specific interactions.** 12 independent simulations were performed for each set of parameters to compute the average droplet number. The star indicates the value used in simulations presented in the main text.

Response:

We thank the reviewer for this comment. We chose the interaction among nucleolar particles to be strong enough to promote phase separation. For simplicity, the strength of specific interactions between the nucleolar particle and chromatin is set the same as that among nucleolar particles.

We performed simulations with different combinations of strength for specific and non-specific interactions between nucleolar particles and chromatin. For each parameter set, we followed the same protocol as in the main text to simulate phase separation. For example, twelve independent 20-million-step-long trajectories were performed, and the droplet number at the end of each simulation trajectory was recorded to compute the averages.

The average number of droplets as a function of specific and non-specific interactions is shown in Fig. R22. We found that as soon as the strength of specific interactions becomes strong enough to promote phase separation, the multi-droplet state emerges. At the strength of $1.4 k_B T$, though droplets begin to emerge in some simulations, there are still significant simulations without phase separation. The average number of droplets in some of the setups is, therefore, less than one. Droplets appear in all simulations at $1.6 k_B T$ and larger values, and many trajectories produce multiple droplets. We decided to use $1.8 k_B T$, which leads to multiple droplets in most simulations. Furthermore, the surface tension of the resulting droplets is comparable to the experimental value as well. The non-specific interaction was chosen as $1.0 k_B T$, but as the phase diagram shows, it has little impact on the results.

We note that in Fig 6 of the main text, we further performed free energy calculations for the specific interaction strength of 1.6 and 2.2. The entropic barrier persists in both cases.

We added more text in the manuscript to motivate our choice of interaction strength and included the above figure in the SI as Fig. S1.

While our results are relatively robust with respect to the strength of these interactions (Fig.S1), we found that, with the chosen values, the surface tension of simulated droplets compares favorably to experimental values for nucleoli (see Supplementary Materials).

Comment 7: *The model assumes purely thermally driven interactions and equilibrium thermodynamics, yet there are several experimental observations that show direct involvement of active processes in life of nuclear bodies, their formation via liquid-liquid phase separation, interactions with the genome as well as the nucleoplasm in vivo (Berry et al, PNAS, 2015, Falahati et al, PNAS, 2017, Caragine et al, eLife, 2019, Kim et al, J. Cell. Sci., 2019). Moreover, in the last two of these studies a correlated motion of nuclear bodies (nucleoli and speckles) was observed in the context of such activity. How do authors reconcile the non-equilibrium nature of the system with their model? And how do their observed correlated motions of the droplets relate to those observed in vivo?*

Response:

We want to point out that the interactions among chromosomes in our model were derived from Hi-C data, which directly incorporates the impact of non-equilibrium activities on genome organization. Therefore, while we presented the results in an equilibrium framework, the interactions and temperature should be viewed as effective values. A large body of literature supports using effective equilibrium models to approximate steady-state distributions of non-equilibrium systems.

The effective equilibrium nature of our model has been clarified in the newly added text in the *Methods Section* as quoted below

We note that the energy unit ($k_B T$) in our model should be viewed as an effective temperature instead of strictly the biological value (300 K). Since the nucleus is a non-equilibrium system with constant perturbations from molecular motors and chemical reactions, the ensemble of genome organizations collected over a population of cells as in Hi-C experiments is unlikely to be in thermodynamic equilibrium. However, non-equilibrium distributions can often be well approximated by renormalized Boltzmann distributions with effective potentials and temperatures [cite]. It is these effective quantities that we inferred from Hi-C contact maps to describe the interactions among chromosomes.

Within the same spirit, in many cases, non-equilibrium activities found in nuclear bodies may be approximated with effective equilibrium models as well. For example, transcriptional activities may effectively tether nuclear bodies toward chromatin, and their role on phase separation could be modeled with attractive interactions between the two.

Figure R23: **Cross-correlation between the displacement vectors of the two droplets.** See text for a detailed definition of the correlation function.

Of course, we agree that other non-equilibrium activities that directly contribute to the formation of nuclear bodies and liquid-liquid phase separation may be beyond the capacity of our modeling effort here. However, the importance of non-equilibrium activities does not necessarily mean that they will outweigh any thermodynamic contributions from passive interactions. They could be complementary to each other, and both play a role in stabilizing the multi-droplet state.

We revised the *Introduction Section* to point out the importance of other complementary mechanisms for stabilizing the multi-droplet state.

However, several observations of nuclear bodies appear to defy predictions from classical nucleation and phase separation theories. In particular, these theories predict the thermodynamic equilibrium to consist of a single condensate that minimizes the surface energy [cite]. On the other hand, multiple nucleoli ($\sim 2-5$) can stably coexist in the same nucleus [cite], as can paraspeckles [cite] and nuclear speckles [cite]. The exact number of nuclear bodies is sensitive to various factors, including cell volume [cite] and nuclear lamina composition [cite]. It has been proposed that non-equilibrium activities can dynamically alter protein-protein interactions to stabilize the multi-droplet state [cite]. In addition, the chromatin network may suppress droplet coarsening through mechanical frustration as well [cite].

We further emphasized the importance of directly incorporating non-equilibrium effects in future studies for a better understanding of nuclear body formation in the revised *Discussion Section* of the manuscript. The relevant text is quoted below.

Finally, it's worth mentioning that the nucleus is a non-equilibrium system, and ac-

tive processes could contribute to the stability of the multi-droplet state as well [cite]. For example, enzymes such as kinase could add post-translational modifications to condensate proteins and regulate their ability in engaging multivalent interactions. Explicitly modeling the active processes within the simulation framework outlined here may be necessary to account for the complementary mechanisms and provide a more comprehensive understanding of in vivo phase separation and nuclear body formation.

To study the correlation between the two droplets, we computed the cross-correlation function of their displacement vector $\vec{v}(t) = \vec{r}(t + \delta t) - \vec{r}(t)$, i.e., the velocity. The correlation function at a time separation of Δt is then computed as $C_v^{\delta t}(\Delta t) = \langle \vec{v}_1(t) \cdot \vec{v}_2(t + \Delta t) \rangle$, where the angular brackets indicate ensemble averaging. Indices 1 and 2 indicate the two individual droplets. $\vec{r}_1(t)$ and $\vec{r}_2(t)$ represent the center of mass position of two droplets at time t . As shown in Fig. R23, the cross-correlation functions $C_v^{\delta t}(\Delta t)$ for different values of δt adopt positive values at $\Delta t = 0$, supporting a positive correlation between the two droplets. This correlation arises as the droplets are tethered to the same chromatin network, the movement of which will drag both droplets concomitantly. The decay to negative values at intermediate timescales before bouncing back to zero indicates the chromatin network's viscoelastic property. Movements of the droplets will compress the network, which will push back later to reverse the droplets' trajectory. Similar dynamical behaviors have indeed been observed in motions of genomic loci measured by live-cell imaging [Khanna et al, Nat Comm, 2019].

We could not find similar experimental measurements for a quantitative/qualitative comparison. It would be interesting for future experiments to validate or falsify this prediction. We note that our estimation here only provides a base correlation under the equilibrium model. It is possible that, as the reviewer mentioned that, non-equilibrium activities contribute to the correlation between nucleoli as well and warrant further investigation.

Comment 8: *The explicit description of the free energy profile for the system containing both the genome and the droplets is not shown. So, it is not clear, how this entropic barrier connects to the role of chromatin in the system.*

Response:

We are not clear on the comment regarding “the free energy profile for the system containing both the genome and the droplets is not shown”. The free energy profile shown in Fig. 3 was computed from the probability distributions, $P(R_g)$, of the collective variable as $F(R_g) = -k_B T \ln P(R_g)$. The probability distributions can be estimated using samples collected from molecular dynamics simulations. We used the umbrella sampling and the weighted histogram technique to compute the free energy across a wide range beyond thermal fluctuation. Since the simulations include both the genome and the nucleolar particles, the contribution of the chromatin network to the free energy profile was accounted for implicitly in the probability distributions.

The reaction coordinate, radius of gyration (R_g), only included coordinates of the nucle-

olar particles. Notably, as shown in Fig. 4A, the probability of configurations collected from the barrier region committing to either the mono-droplet or two-droplet state is close to 0.5, which is the value expected for a perfect transition state [Du et al., JCP, 1998]. Therefore, R_g serves as a reasonable reaction coordinate to study the mechanism of the droplet coalescence. The fact that coordinates of the chromatin network are not needed to define the reaction coordinate suggests that chromatin plays a passive role for droplet coalescence, i.e., as a viscoelastic medium.

The most direct evidence for the contribution of chromatin network to the entropic barrier is shown in Fig. S3. In these independent sets of simulations, the connectivity between genomic segments is removed. For example, chromatin is dissolved from connected polymers to separate particles. Dynamical simulations of the dissolved network resulted in mono-droplet states. Furthermore, free energy calculations showed that the entropic barrier disappears in this system as well. Since the only perturbation to the system is removing the chromatin network, these results support its essential role in giving rise to the barrier.

We revised the main text to further clarify our reasoning for the importance of chromatin network.

Simulations of the full system with both chromosomes and nucleolar particles were used to compute the free energy profile. Therefore, the impact of the chromatin network was accounted for implicitly even though it was not included in the definition of R_g .

Comment 9: *Authors are wording their work as if it generally applies to all nuclear bodies, but their model accounts for existence of NADs, allowing for protein-chromatin interactions, which are an exclusive feature of nucleoli. Hence, a generalization of this model to other nuclear bodies, which have no specific interactions with NADs (or other genomic sequences), is unclear.*

Response:

We thank the reviewer for this comment, and agree that interactions between nuclear bodies and chromatin are not well characterized. While nucleoli are known to bind chromatin regions encoding rDNA, other nuclear bodies are less clear. We revised the main text in numerous places to explicitly emphasize our focus on nucleoli.

OLD text

Using a diploid human genome model parameterized with chromosome conformation capture (Hi-C) data, we studied the thermodynamics and kinetics of droplet formation inside the nucleus.

NEW text

Using a diploid human genome model parameterized with chromosome conformation capture (Hi-C) data, we studied the thermodynamics and kinetics of nucleoli formation.

OLD text

Therefore, protein-chromatin interactions arrest phase separation in multi-droplet states and may drive the variation of nuclear body numbers across cell types.

NEW text

Therefore, the chromatin network supports a nucleation and arrest mechanism to stabilize the multi-droplet state for nucleoli and possibly for other nuclear bodies.

OLD text

We carried out molecular dynamics simulations to investigate LLPS in the nucleus with a computational model that includes protein particles and the chromatin network.

NEW text

We carried out molecular dynamics simulations to investigate nucleoli formation with a computational model that explicitly considers nucleolar particles, the chromatin network, and the interactions between the two.

OLD text

Leveraging a recently introduced computational model for the diploid human genome [cite], we studied the impact of the chromatin network on phase separation inside the nucleus.

NEW text

Leveraging a recently introduced computational model for the diploid human genome [cite], we studied the impact of the chromatin network on nucleoli formation.

OLD text

We introduced additional coarse-grained particles to model protein molecules.

NEW text

We introduced additional coarse-grained particles to model molecules that make up the nucleoli.

OLD text

We studied the role of protein-chromatin interactions for phase separation inside the nucleus.

NEW text

We modeled the dynamical process of phase separation that drives nucleoli formation in the presence of the chromatin network.

However, at least for speckles, recent literature also supports the presence of specific interactions. We added the following text to the *Discussion Section* of the manuscript to comment on the generality of our study to other nuclear bodies.

Specific interactions between nucleolar particles and chromatin are crucial for the nucleation step and the emergence of the multi-droplet state (Fig. S1). As such interactions may be present for other nuclear bodies, a similar mechanism could contribute to their formation as well. For example, several recent studies support the close contact between a subset of chromatin and speckles with distances at ~ 100 nm or less using techniques based on high throughput sequencing [cite] or imaging [cite]. Furthermore, these speckle-associated chromatin regions are largely conserved across cell-types [cite], supporting the presence of non-random mechanisms for their maintenance. Close contacts with chromatin are likely stabilized by RNA molecules that are known to localize at speckle periphery [cite]. Non-coding RNAs have also been found in paraspeckles and PML bodies to mediate their interactions with chromatin [cite].

We revised the *Introduction* of the main text to further clarify the nature of interactions between nuclear bodies and chromatin. The updated text is quoted below

Nuclear bodies and chromatin are also known to form attractive interactions [cite], further complicating phase separation inside the nucleus beyond mechanical stress. For example, the upstream binding factor (UBF), which is a DNA binding protein and a key component of nucleoli, is known to recognize ribosomal DNA (rDNA) repeats to seed the rapid formation of nucleoli after cell division [cite]. Correspondingly, rDNA and other chromosome segments, which are collectively noted as nucleolus-association domains (NADs) [cite], can be seen inside and adjacent to nucleoli [cite]. Paraspeckles [cite] and speckles [cite] have been found in spatial proximity with chromatin as well. In addition to proteins, nuclear bodies can harbor non-coding RNA to contact chromatin either by recruiting intermediate protein molecules or by forming RNA-DNA duplex or triplex [cite]. Since chromatin forms a viscoelastic network spanning the nucleus [cite], its interactions with nuclear bodies could impact the thermodynamics and kinetics of phase separation.

Comment 10: *Fig. S4 A – there is twice a typo in the cartoon: “Large Radius of Gryation”.*

Response:

The figure has been updated to correct the typo.

REVIEWERS' COMMENTS

Reviewer #1 (Remarks to the Author):

I appreciate the authors' efforts in performing additional simulations and comparing them with the recent experimental data; it certainly improved the manuscript. The authors honestly discussed the limitations of their model, which nevertheless is useful and provides a phenomenological understanding of chromatin folding.

Reviewer #2 (Remarks to the Author):

This revised version of the manuscript by Qi and Zhang has been extensively improved. The authors have satisfactorily addressed all my previous comments. I am particularly impressed at the effort they made to address stress/pressure level in the droplets, and the connection they make to wetting of chromatin. This is a convincing and valuable addition to an already good paper. In my opinion, this article can be published as is in Nature Communication.

Reviewer #3 (Remarks to the Author):

The authors have addressed all of my comments and answered most of my concerns. Indeed, the revised manuscript has strongly improved. There are still some minor issues that need to be resolved before I can recommend publication in Nature Communication:

1. In the abstract the authors refer to correlated motion of nucleoli and chromatin – as droplets coalesce, the chromatin network becomes increasingly constrained. I think the use of word “correlated” may be a bit misleading here as that can have a precise mathematical meaning, yet the authors are purely stressing the close relation/link between the two systems – as one coalesces, the other one becomes constrained since both of them co-exist in the confinement of the nucleus. So “closely related/linked” may be more appropriate.

2) In response to my earlier comment # 9, throughout the revised manuscript the authors now refer to the droplets as nucleoli due to their attachment to the NADs. This change should be reflected also in the following places in the manuscript:

a) The title should also reflect this change and focus on the nucleoli.

b) Similarly, since the revised manuscript now focuses on the nucleoli - the comparison of the coarsening kinetics of nucleoli in simulations should be made to experiments measuring nucleolar coarsening in vivo, instead of optodroplets in vivo (Lee et al) as it is currently done on page 13. The optodroplets are not associated with chromatin and can freely move, hence their behavior may be different. This comparison should be done against experiments measuring nucleolar coarsening in vivo.

3) The authors refer to biological temperature value as 300K, which would be 27C. I assume, they meant 310K, which is the physiological temperature of 37C.

Title: Chromatin Network Retards Droplet Coalescence

Manuscript ID: NCOMMS-21-07939

Authors: Yifeng Qi and Bin Zhang

We want to thank all the referees again for their time and effort in reviewing the manuscript. In the following, we provide point-to-point responses to the comments of reviewer #3.

RESPONSE TO REFEREE 3

Comment 0: *The authors have addressed all of my comments and answered most of my concerns. Indeed, the revised manuscript has strongly improved. There are still some minor issues that need to be resolved before I can recommend publication in Nature Communication:*

Response:

We appreciate the reviewer's positive assessment of the paper, and we thank him/her for the detailed suggestions and comments.

Comment 1: *1. In the abstract the authors refer to correlated motion of nucleoli and chromatin – as droplets coalesce, the chromatin network becomes increasingly constrained. I think the use of word “correlated” may be a bit misleading here as that can have a precise mathematical meaning, yet the authors are purely stressing the close relation/link between the two systems – as one coalesces, the other one becomes constrained since both of them co-exist in the confinement of the nucleus. So “closely related/linked” may be more appropriate.*

Response:

We thank the reviewer for the comment and have replaced “correlated” with “coupled”.

Comment 2.a: *2) In response to my earlier comment # 9, throughout the revised manuscript the authors now refer to the droplets as nucleoli due to their attachment to the NADs. This change should be reflected also in the following places in the manuscript: a) The title should also reflect this change and focus on the nucleoli.*

Response:

We have revised the title to “Chromatin Network Retards Nucleoli Coalescence”.

Comment 2.b: *b) Similarly, since the revised manuscript now focuses on the nucleoli - the comparison of the coarsening kinetics of nucleoli in simulations should be made to experiments measuring nucleolar coarsening in vivo, instead of optodroplets in vivo (Lee et al) as it is currently done on page 13. The optodroplets are not associated with chromatin and can freely move, hence their behavior may be different. This comparison should be done against experiments measuring nucleolar coarsening in vivo.*

Response:

We thank the reviewer for this comment. The only in vivo data for the coarsening kinetics of nucleoli we could find is from [PNAS, 112:E5237-E5245 (2015)] done by Brangwynne and colleagues. In this paper, the authors examined the dynamic scaling of nucleolar

Figure R1: **Mean integrated intensity of nucleoli over time.** The left panel was adapted from Figure 1B of Ref [PNAS, 112:E5237-E5245 (2015)]. The authors fitted data with $t \leq 8\text{min}$ to obtain a scaling exponent of 0.33. However, our independent fitting using only data with $t \leq 5\text{min}$ results in an exponent of 0.18. The value of n , which denotes the scaling of the nucleolar size with respect to the time, was calculated as one-third of the fitting coefficient for the mean integrated intensity as a function of time: $\langle I \rangle \sim V \sim R^3 \sim t^{3n}$. The underlying assumption is that the intensity is proportional to the nucleolar volume which is proportional to R^3 . Our visual inspection of the experimental data in the left panel suggests that the jump occurring at 5min better separates the data into two regimes. From the left panel, it is also evident that experimental data is rather noisy, rendering a robust extraction of the exponent difficult.

coarsening by plotting the average intensity of nucleoli as a function of time. The measurements were quite noisy with large error bars, rendering an accurate estimation of the exponent difficult. Furthermore, there are distinct regimes of dynamic scaling behaviors at different time intervals (Figure R1, Left). For example, the authors obtained a value of 0.33 ± 0.14 , but our estimation using short-time data with $t \leq 5\text{min}$, rather than using $t \leq 8\text{min}$ as done in the manuscript, resulted in an exponent of 0.18 (Figure R1). Considering the statistical noise, we conclude that this number agrees reasonably with our simulation value.

We revised the manuscript to add the following text for comparing with in vivo data.

The scaling exponent for nucleolar coarsening in vivo is also in good agreement with the simulated value when considering short time kinetics before 5 mins [cite].

Comment 3: 3) The authors refer to biological temperature value as 300K, which would be 27C. I assume, they meant 310K, which is the physiological temperature of 37C.

Response:

We thank the reviewer for this comment and have revised the temperature value to 310K in the text.